# An MPER antibody neutralizes HIV-1 using germline features shared among donors

Lei Zhang[1,22,25], Adriana Irimia[2,3,4,25], Lingling He[1,2], Elise Landais[1,3,5], Kimmo Rantalainen[2], Daniel P. Leaman[1], Thomas Vollbrecht [6], Armando Stano[1], Daniel I. Sands [1], Arthur S. Kim [1,23], IAVI Protocol G Investigators, Pascal Poignard[1,3,5,24], Dennis R. Burton [1,3,4,7], Ben Murrell[6,8], Andrew B. Ward [2,3,4], Jiang Zhu[1,2,26*], Ian A. Wilson [2,3,4,9,26*] & Michael B. Zwick[1,26*]

The membrane-proximal external region (MPER) of HIV-1 envelope glycoprotein (Env) can be targeted by neutralizing antibodies of exceptional breadth. MPER antibodies usually have long, hydrophobic CDRH3s, lack activity as inferred germline precursors, are often from the minor IgG3 subclass, and some are polyreactive, such as 4E10. Here we describe an MPER broadly neutralizing antibody from the major IgG1 subclass, PGZL1, which shares germline V/D-region genes with 4E10, has a shorter CDRH3, and is less polyreactive. A recombinant sublineage variant pan-neutralizes a 130-isolate panel at 1.4 µg/ml ($IC_{50}$). Notably, a germline revertant with mature CDR3s neutralizes 12% of viruses and still binds MPER after DJ reversion. Crystal structures of lipid-bound PGZL1 variants and cryo-EM reconstruction of an Env-PGZL1 complex reveal how these antibodies recognize MPER and viral membrane. Discovery of common genetic and structural elements among MPER antibodies from different patients suggests that such antibodies could be elicited using carefully designed immunogens.

[1] Department of Immunology and Microbiology, The Scripps Research Institute, La Jolla, California 92037, USA. [2] Department of Integrative Structural and Computational Biology, The Scripps Research Institute, La Jolla, California 92037, USA. [3] International AIDS Vaccine Initiative Neutralizing Antibody Center and the Collaboration for AIDS Vaccine Discovery, The Scripps Research Institute, La Jolla, California 92037, USA. [4] Scripps Consortium for HIV/AIDS Vaccine Development, The Scripps Research Institute, La Jolla, California 92037, USA. [5] International AIDS Vaccine Initiative, New York, New York 10004, USA. [6] Department of Medicine, University of California, San Diego, California 92093, USA. [7] Ragon Institute of Massachusetts General Hospital, MIT and Harvard, Cambridge, Massachusetts 02114, USA. [8] Department of Microbiology, Tumor and Cell Biology, Karolinska Institutet, Stockholm, Sweden. [9] Skaggs Institute for Chemical Biology, The Scripps Research Institute, La Jolla, California 92037, USA. [22] Present address: CTK Biotech, Inc., 3855 Stowe Drive, Poway, California 92064, USA. [23] Present address: Departments of Medicine, Pathology and Immunology, Washington University School of Medicine, St. Louis, Missouri 63110, USA. [24] Present address: Institut de Biologie Structurale, Université Grenoble Alpes, Commissariat a l'Energie Atomique, Centre National de Recherche Scientifique and Centre Hospitalier Universitaire Grenoble Alpes, 38044 Grenoble, France. [25] These authors contributed equally: Lei Zhang, Adriana Irimia. [26] These authors jointly supervised: Jiang Zhu, Ian A. Wilson, Michael B. Zwick. A full list of Consotium members appears at the end of the paper. *email: jiang@scripps.edu; wilson@scripps.edu; zwick@scripps.edu

A key goal in HIV vaccine design is to elicit broadly neutralizing antibodies (bnAbs)[1]. Most bnAbs to HIV-1 have been cloned from elite donors whose plasma shows broad neutralizing activity. These bnAbs target six distinct sites on the HIV-1 envelope glycoprotein (Env) spike, including the CD4-binding site (CD4bs), V2 apex, N332/V3 base supersite, silent face, gp120-gp41 interface (including fusion peptide), and membrane-proximal external region (MPER). As bnAbs arise from complex affinity maturation pathways, efforts are underway to dissect the structural and genetic bases of bnAb function to uncover common elements that can simplify vaccine design[2].

MPER bnAbs show outstanding breadth, neutralizing up to >98% primary isolates, but have uncommon features[3]. MPER bnAbs are often from the IgG3 subtype, which has caused speculation that eliciting these bnAbs involves certain B-cell subsets, class-switching, or a specific hinge region[4,5]. Although 2F5, 4E10, 10E8, DH511, DH517, VRC46, and VRC43.01 are IgG3s, the MPER bnAb VRC42 was isolated as an IgG1 from the same subject as the latter two bnAbs[5]. Notably, long heavy complementarity-determining region (CDR) H3 loops with aromatic residues at the tip facilitate bnAb binding to the hydrophobic MPER and nearby membrane[6]. However, B-cell receptors (BCRs) with long and hydrophobic CDRH3s tend to be downregulated during B-cell ontogeny[7] and some MPER bnAbs, e.g., 4E10, are mildly polyreactive[8]. Further, 4E10 knock-in mice exhibit B-cell tolerance via clonal deletion and anergy[9]. BnAbs 10E8 and DH511 were recently shown to recognize a similar epitope as 4E10 with less polyreactivity and higher potency[10,11], but key information is missing on the precise antigens and mechanisms that drove their evolution.

The hydrophobic MPER is often truncated from Env constructs to render soluble gp140 trimers[12]; thus, the MPER has been commonly studied in isolation. MPER peptide, $N_{671}WFDITNWLWYIK_{683}$, adopts a mainly α-helical conformation with $W_{672}–D_{674}$ in a $3_{10}$ helix when bound to 4E10 and is fully helical when bound to 10E8 and DH511[11,13]. However, MPER peptides constrained as an α-helix have not elicited nAbs[3]. One issue is that the membrane can hinder antibody access to the MPER on the virus[6]. Further, cryogenic electron microscopy (cryo-EM) reconstructions have revealed interaction of 10E8 with N-linked glycans on membrane-extracted Env at positions 88 and 625[14]. Thus, elicited antibodies should accommodate membrane and adjacent glycans on the Env trimer. MPER accessibility increases transiently when Env binds to CD4 receptor, just prior to co-receptor binding and virus entry into host cells, but structural details of this transient state are lacking.

Recently, vaccine design has focused on targeting common elements among certain bnAb precursors. For example, VRC01-class CD4bs antibodies typically use germline gene $V_H1-2$, for which specific immunogens have been designed[15]. Germline-encoded residues important for Env recognition by different V2 apex bnAbs have also been identified[16], whereas other bnAb precursors recognize a transmitted-founder (T/F) Env[17]. Germline revertants of many bnAbs do not bind to Env, although some somatic hypermutations (SHM) are dispensable with 4E10 and 10E8[18,19]. Interestingly, a recently described 4E10-like bnAb, VRC42, plus two other MPER bnAb lineages with limited SHM, were elicited by a single T/F Env[5].

Here we report on MPER bnAb PGZL1 and its lineage variants, which, along with 4E10 and VRC42, share the usage of $V_H1-69$ and $V_K3-20$ V-region germline genes, as well as D-region allele D3-10. PGZL1 is an IgG1 that has a relatively short CDRH3 with limited hydrophobicity and is less polyreactive than 4E10; PGZL1 germline revertants can still bind Env and neutralize HIV. Thus, 4E10-like antibodies may be more probable to elicit in humans than previously thought and, therefore, provide clues for vaccine design.

## Results

### Identification of a 4E10-like antibody from an African donor.
Plasma from visit 2 (out of a total of six visits) of an HIV-1-infected South African donor, PG13, in the IAVI Protocol G cohort[20] was tested for neutralization against HIV-2 (HIV-1 MPER) chimeric viruses. The titer was 1:6400 ($ID_{50}$) against 4E10-sensitive chimeras C1 and C4, but the 2F5-sensitive C3 chimera was not neutralized, suggesting a 4E10-like antibody (Fig. 1a and Supplementary Table 1). The plasma also neutralized five of a six-virus panel (Fig. 1a). MPER peptide partially blocked plasma neutralization of primary isolate Du156.12 and the C1 chimera (Supplementary Table 1). MPER-positive B cells from PG13 visit 5 were then sorted by fluorescence-activated cell sorting (FACS; see Methods), and the resulting heavy-chain (HC) and light-chain (LC) variable regions were cloned into IgG vectors. Antibody PGZL1 bound to full-length MPER and a 4E10-specific peptide, but not to a 2F5-specific peptide (Fig. 1b).

DNA sequencing revealed PGZL1 was from subclass IgG1, whereas most MPER bnAbs are IgG3s (Supplementary Table 2). Phylogenetic analysis showed sequence homology between PGZL1, VRC42.01, and 4E10, due to shared germline genes $V_H1-69$, $V_K3-20$, and $D_H3-10*01$ (Fig. 1c and Supplementary Table 2). PGZL1 has a high degree of SHM (20.9% nucleotides for HC and 12.6% for LC; Fig. 1e), and five HC and two LC mutations match those in 4E10 (Supplementary Fig. 1). PGZL1, VRC42.01, and 4E10 use different J-genes. The PGZL1 CDRH3 contains 15 residues, 3 residues shorter than 4E10, and equal in length to VRC42.01. The three CDRH3s share some amino acids not only from the same D-gene, e.g., $W_{99}$, $G_{100a}$, but also at positions $E_{95}$, $G_{96}$, $G_{98}$, $K_{100b}$, $P_{100c}$, and $A_{100f}$, which may have arisen from N-additions or SHM (Supplementary Fig. 1).

### Broad HIV neutralization by PGZL1 lineage antibodies.
We tested PGZL1 against a six-virus panel and found that it neutralized five viruses with an $IC_{50}$ of 5.8 µg/ml, which was sixfold less potent than 4E10 (Fig. 1a). To identify more potent PGZL1 variants, we analyzed BCR transcripts of donor PG13 B cells taken from three visits (visit 2, 4, and 6) in a short 9-month span using a next-generation sequencing (NGS) pipeline[21] (Supplementary Table 3). The PG13 repertoire profiles were similar among visits, with PGZL1 germline gene $V_H1-69$ accounting for 3.9–7.2% of the repertoire, whereas $V_H4-34$ and $V_H4-59$ composed up to 13% (Supplementary Fig. 2). Two-dimensional identity/divergence analysis (Fig. 1d) and CDRH3-based lineage tracing showed two groups of PGZL1-related HCs: one group formed an island with full-length identity of 85–100% and the other was created by a long stretch of up to 80% identity, suggesting a sublineage related to PGZL1. As earlier PG13 peripheral blood mononuclear cells (PBMCs) and plasma samples were depleted or unavailable, the time of lineage division could not be determined. Overall, the PGZL1 lineage appears to be highly evolved, showing similar patterns to the VRC01-class bnAbs that often diverge into many sub-lineages[22].

For functional studies, a clustering analysis of CDRH3s with at least 80% identity allowed a broad selection of 27 HCs from the PGZL1 sub-lineages (Supplementary Fig. 3a and Supplementary Table 4). CDRH3 remained conserved in length and sequence, perhaps due to affinity maturation against the conserved MPER. We paired the HCs with the PGZL1 LC and tested these antibodies against Du156.12. The best neutralizers, H4 and H8 from Visit 2 (Supplementary Table 4), had two- to fourfold lower $IC_{50}$s than PGZL1 (Supplementary Fig. 3b). Of note, H4 and H8 seemed to come from a distinct sublineage having low sequence identity (≤83%) with respect to PGZL1 HC (Fig. 1d). H4 and H8 HCs were then paired with 10 similarly chosen NGS

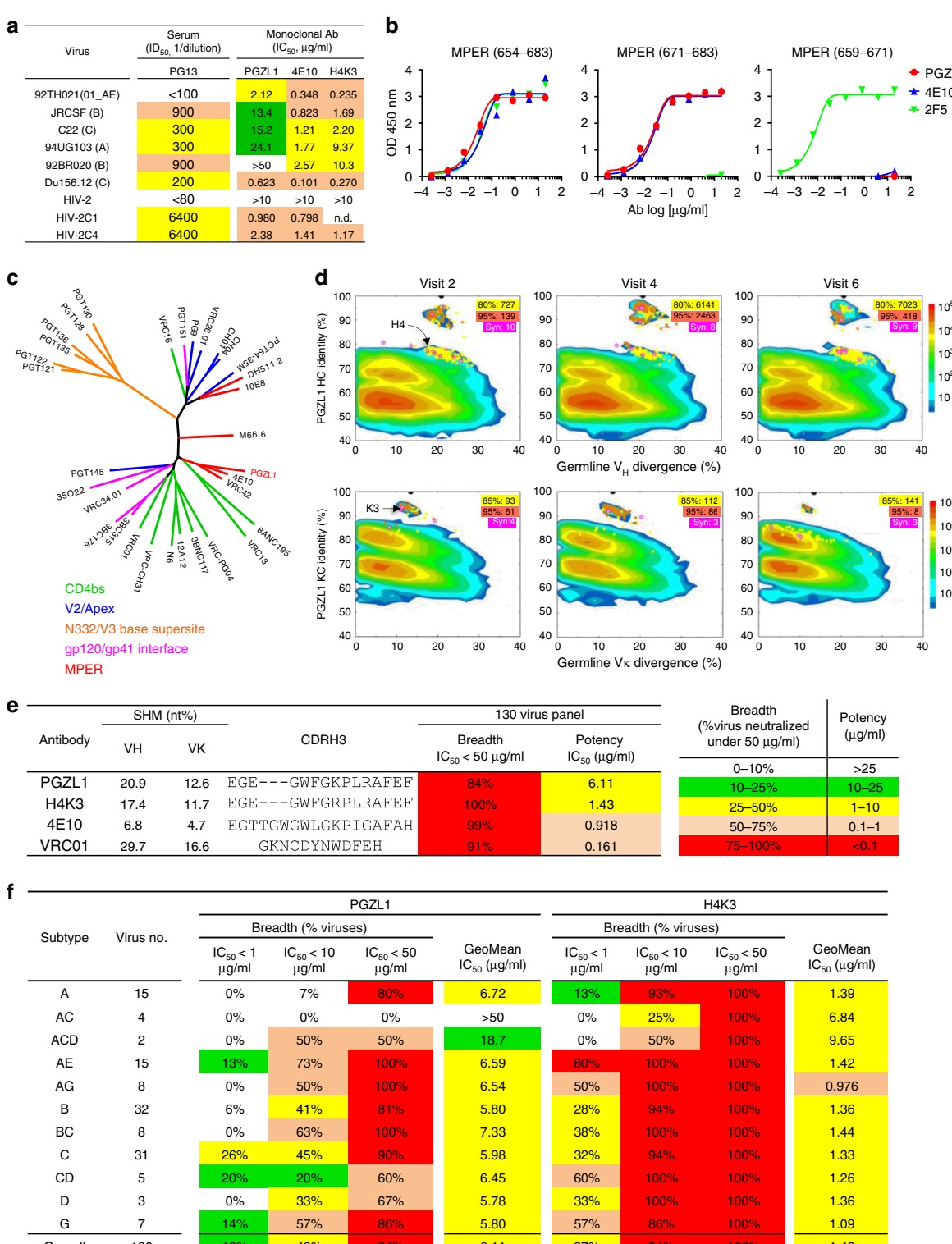

**Fig. 1** Properties of an MPER-targeted bnAb. **a** Neutralization of HIV-1 six-virus panel and HIV-2 (HIV-1 MPER) chimeras by PG13 plasma and monoclonal antibodies PGZL1, 4E10, and H4K3. **b** ELISA binding of PGZL1 to MPER peptides, using 4E10 and 2F5 as controls. **c** Maximum likelihood (ML) tree of HC variable regions of described bnAbs, colored by Env specificity. **d** Divergence/identity analysis of donor PG13 antibody repertoire over three visits in 9 months. NGS-derived antibody chains are plotted as a function of sequence identity to PGZL1 and divergence from their putative germline genes. Colors indicate sequence density. Sequences with a CDR3 identity of ≥80/85% (HC/KC) and with a CDR3 identity of ≥95% are shown as yellow and orange dots on the 2D plots, with the number of sequences highlighted in yellow and orange shades, respectively. Sequences bioinformatically selected for synthesis are shown as magenta stars on the 2D plots, with the number of sequences (Syn) highlighted in magenta shade. **e**, **f** Neutralization breadth and potency of PGZL1 and H4K3 against a 130-virus panel (**e**) and the same data as in **e** but subdivided by HIV subtype (**f**).

variant LCs (Supplementary Table 4). PGZL1 antibodies, H4K3 and H8K3, which contain the LC K3 from donor Visit 2, neutralized Du156.12 and 92TH021 with a further two- to fourfold decrease in $IC_{50}$s (Supplementary Fig. 3c). PGZL1.H4K3 (hereafter H4K3) was chosen for more detailed characterization. H4K3 has less SHM than PGZL1, i.e., 17.4%/11.7% vs. 20.9%/ 12.6% at the nucleotide level for HC/LC, respectively, consistent with H4K3 being sampled 5 months prior to PGZL1.

We used a larger panel of HIV-1 primary isolates to examine breadth and potency of PGZL1 and H4K3. Notably, H4K3 neutralized 100% viruses at $\leq 50$ μg/ml ($n = 130$), with an $IC_{50}$ of 1.43 μg/ml (Fig. 1e, f and Supplementary Table 5). 4E10 gave similar results, neutralizing 99% isolates with an $IC_{50}$ of 0.92 μg/ ml. PGZL1 neutralized 84% viruses, with an $IC_{50}$ of 6.11 μg/ml. The $IC_{50}$s of PGZL1 and H4K3 are tightly correlated ($r = 0.82$, $p < 0.0001$), suggesting similar neutralization mechanisms (Supplementary Fig. 3d). VRC01 neutralized 91% viruses with an $IC_{50}$ of 0.161 μg/ml. Overall, H4K3 is exceptionally broad and equipotent with 4E10, which has been reported to protect against SHIV challenge in monkeys[23].

**PGZL1 germline revertant binds the MPER and neutralizes HIV.** Germline revertants of 4E10[24] and 10E8[19] reportedly do not neutralize HIV-1. We reverted the V-region of PGZL1 to the most homologous germline alleles, i.e., $V_H1$-69*06, $V_K3$-20*01, while leaving the CDR3s unchanged, thus creating PGZL1 gVmDmJ; in a second antibody PGZL1 gVgDgJ, putative reversions were also made to $D_H3$-10*01 and $J_H3$*01 (Fig. 2a and Supplementary Fig. 4a). Surprisingly, PGZL1 gVmDmJ and gVgDmJ still bound the MPER peptide, but with a 6- to 12-fold increase in $EC_{50}$ (Fig. 2b); DJ-reverted PGZL1 gVgDgJ also bound, albeit with a 369-fold increase in $EC_{50}$. Analogous revertants 4E10 gVmDmJ, 4E10 gVgDgJ, and 10E8 gVmDmJ showed little or no such binding as previously reported (Fig. 2b)[18,19].

Using biolayer interferometry (BLI), PGZL1, PGZL1_gVmDmJ, and PGZL1_gVgDgJ bound with affinities ($K_D$) ranging from 9.9 nM to 64.4 nM and 938 nM, respectively, mostly due to different off-rates (Fig. 2c). We created another mutant, PGZL1 gVgDmJ, to test the effect of changes in $D_H$. PGZL1 gVgDmJ bound the MPER peptide with a $K_D$ of 148 nM, indicating that reversions in both $D_H$ and $J_H$ affect the PGZL1 off-rate. PGZL1 and H4K3 had similar $K_D$s of 9.9 nM and 7.5 nM, respectively, but on- and off-rates of H4K3 to MPER peptide were faster by about 24 and 8%, respectively (Fig. 2c).

To test whether the gVmDmJ germline revertants could bind to the MPER on the cell surface, we created a 293T cell line that displays the MPER and transmembrane domain (MPER-$TM_{654-709}$). The PGZL1 gVmDmJ revertant, but not 4E10 or 10E8 revertants, also stained MPER-TM cells, albeit not as well as mature PGZL1, H4K3, 4E10, and 10E8 (Fig. 2d). We then assessed antibody staining of a cell line that overexpresses Env[25]. We observed moderate staining of high-Env cells by PGZL1 in the absence of sCD4 and strong staining by 4E10, 10E8, and H4K3 (Fig. 2d). Cell surface Env was not stained by the revertants at a concentration of 10 μg/ml (Fig. 2d). Addition of sCD4 enhanced MPER bnAb staining of Env cells, as expected; notably, sCD4 also enabled staining by PGZL1 gVmDmJ and 4E10 gVmDmJ, but not by 10E8 gVmDmJ. Of note, the 4E10 revertant reportedly bound weakly to a deglycosylated MPER-containing gp140[26].

We also assessed binding of PGZL1 variants to the detergent-extracted Env ADA-CM, which is the ADA Env containing trimer stabilizing mutations ΔN139, ΔI140, N142S, I535M, L543Q, K574R, H625N, T626M, and S649A[27]. In a blue-native polyacrylamide gel electrophoresis (BN-PAGE) mobility shift assay, both PGZL1 and H4K3 shifted Env (Fig. 2e). Notably, PGZL1 gVmDmJ and gVgDgJ also shifted Env, but gVgDgJ required a higher concentration and did not reach a binding stoichiometry of three Fabs per trimer. BnAb 10E8 and non-nAb b6 controls showed the predicted gel shift and lack of shift, respectively (Fig. 2e). Similar results were observed using Envs HxB2 and Du156.12 (Supplementary Fig. 4b). Hence, PGZL1 gVgDgJ binds to Env when detergent-extracted from membrane, but not on the cell surface, whereas PGZL1 gVmDmJ binds both solubilized and cell surface Env.

Strikingly, PGZL1 gVmDmJ neutralized 12% of isolates at 50 μg/ml and 28% of viruses at 200 μg/ml in a dose-dependent manner (Fig. 2f, g and Supplementary Table 5). The $IC_{50}$s of PGZL1 and PGZL1 gVmDmJ correlated modestly ($r = 0.52$, $p = 0.0013$) (Supplementary Fig. 3d). We tested viruses especially sensitive to PGZL1 gVmDmJ against additional revertants of PGZL1 and 4E10 (Supplementary Fig. 4). No virus was neutralized by PGZL1 gVgDgJ or 4E10 gVgDgJ at 200 μg/ml (Supplementary Fig. 4c), despite the ability of PGZL1 gVgDgJ, but not 4E10 gVgDgJ, to bind the MPER (Fig. 2b). However, 4E10 gVmDmJ neutralized 92TH021 and CNE8, whereas 4E10 gVmDmJ_$L_{100c}$F, which has the $D_H3$-10*01-encoded $F_{100c}$ and binds MPER ~ 2-fold better than 4E10 gVmDmJ (Fig. 2c), neutralized these isolates more potently in addition to T255 and X1193 (Supplementary Fig. 4c). Thus, ~ 1.4% isolates were neutralized by revertants of PGZL1 and 4E10, which had the D-gene-encoded $F_{100}$ ($L_{100c}$ in 4E10; Supplementary Fig. 1a) in the context of mature CDR3s.

4E10 neutralizes HIV-1 by a slow and/or post-CD4-attachment mechanism, as it can be washed off the virions prior to adding to target cells[10]. We performed washouts with PGZL1 gVmDmJ, using VRC01 as a control that binds directly to Env and resists washout. Neutralization of Du156.12 by PGZL1 gVmDmJ was partially lost on washout but was unchanged with VRC01 (Supplementary Table 6). Notably, PGZL1 gVmDmJ neutralized 928.28, BJOX025000, HxB2, and 92TH021, albeit modestly, following the washout (Supplementary Table 6). Thus, PGZL1 gVmDmJ binds weakly to and neutralizes some primary isolates prior to receptor engagement.

**PGZL1 uses 4E10-like features but has limited polyreactivity.** To dissect PGZL1 activity, we created chimeras of PGZL1 and 4E10, and tested their ability to neutralize 92TH021. Engraftment of 4E10 CDRs H1 and H2 onto PGZL1 produced a modest ~ 2-fold increase in potency (Fig. 3a). Engrafting CDRH3 of 4E10 onto PGZL1 caused a 20-fold increase in neutralization. Substituting the LC or HC of PGZL1 with that of 4E10 produced 3-fold and 10-fold increases in potency, respectively. Thus, PGZL1 becomes more potent using 4E10 LC or HC CDRs, and in particular its CDRH3.

The most prominent difference in CDRH3s of PGZL1 and 4E10 is the additional Thr-Gly-Trp (TGW) motif in 4E10 (Supplementary Fig. 1a). This motif increases CDRH3 hydrophobicity (Supplementary Fig. 1b); thus, this might enhance neutralization via membrane interaction. Of note, a putative variant of VRC42.01, termed VRC42.N1, was identified using plasma proteomics and contains a Glu-Gly-Trp (EGW) insertion at the analogous position in VRC42[5]. Indeed, insertion of TGW improved PGZL1 neutralization roughly threefold, but the fold-change varied among isolates in the six-virus panel, indicating paratope context also affected neutralization. CDRH3s of PGZL1 and 4E10 differ at position 100, which is the $D_H3$-10*01-encoded $F_{100}$ in PGZL1, but somatically mutated to $L_{100c}$ in 4E10. The hydrophobic mutant 4E10 $L_{100c}$F neutralized threefold more potently than wild-type 4E10 (Fig. 3b) and was

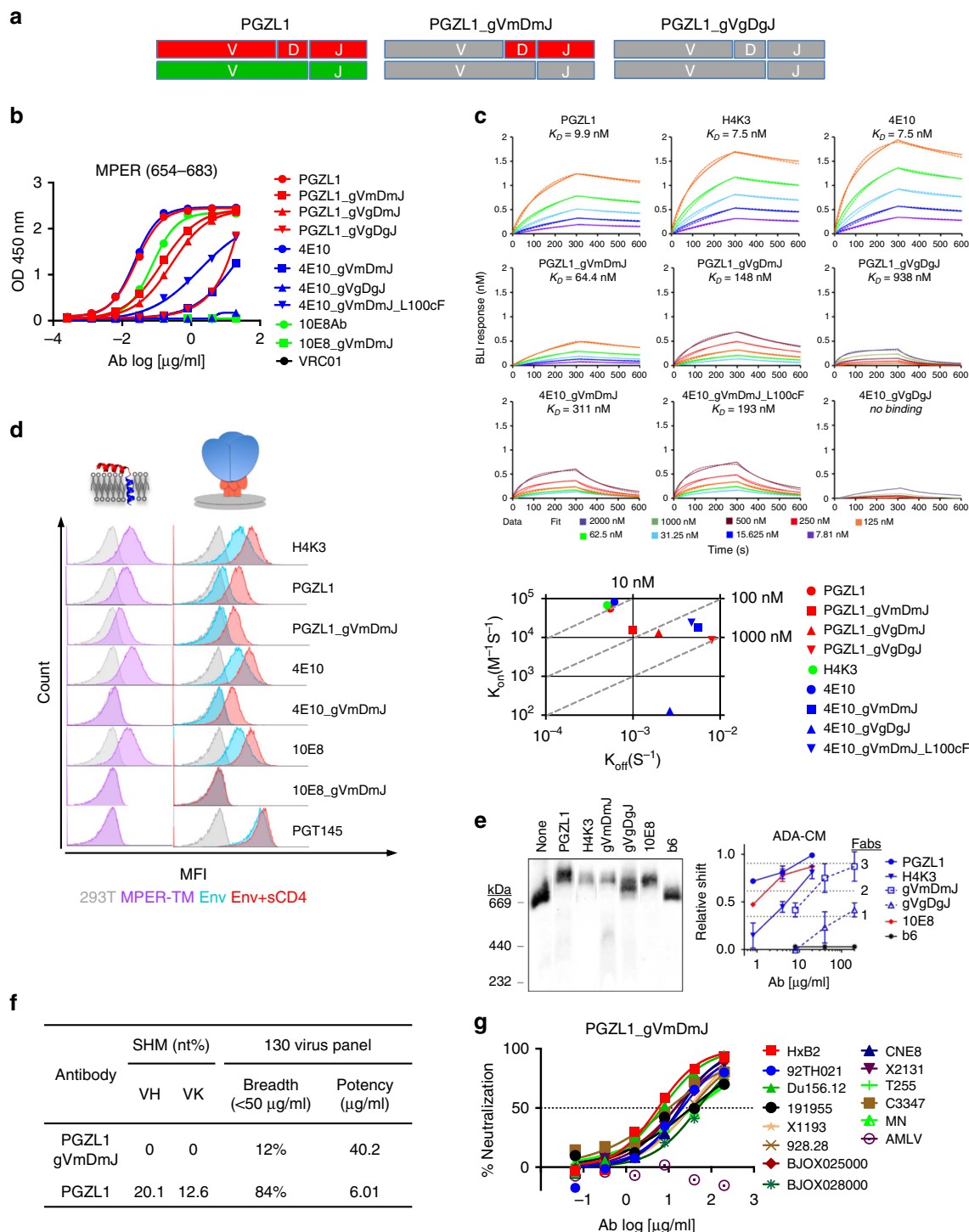

**Fig. 2** Characterization of germline-reverted antibody PGZL1 gVmDmJ. **a** Cartoon of mature PGZL1 $V_H$ (red; top) and Vκ (green; bottom) subdivided by V, D, and J regions, and germline reversions (gray) to create PGZL1 gVmDmJ (middle) and PGZL1 gVgDgJ (right). **b** ELISA binding of PGZL1 germline revertants to MPER peptide, using analogous 10E8 and 4E10 controls. **c** BLI-binding kinetics of PGZL1 variants to immobilized MPER peptide (top panels). 4E10 variants were also used for comparison. $k_{on}$ and $k_{off}$ of antibodies are shown on a scatter plot with affinity constant, $K_D$, as dashed lines (bottom). **d** Cells expressing MPER-TM (purple histograms, left) were stained in flow cytometry by mature and germline-reverted antibodies at 2 μg/ml. HIV Env (right) in the presence and absence of soluble CD4 (sCD4; red and blue histograms, respectively) were stained by mature and germline-reverted antibodies at 2 and 10 μg/ml, respectively. **e** BN-PAGE Env mobility shift assay. HIV-1 virions were incubated with Fab PGZL1 and H4K3 (20 μg/ml), or PGZL1 gVmDmJ and gVgDgJ (200 μg/ml). 10E8 (20 μg/ml) and non-neutralizing antibody b6 (200 μg/ml) were used as positive and negative controls, respectively. Relative shift and stoichiometry of Fab to Env was quantified. The error bars represent the SD of $n = 2$ biologically independent experiments. **f** Neutralization potency and breadth of PGZL1 gVmDmJ against a 130-virus panel of HIV-1 in TZM-bl assay at 200 μg/ml. **g** Neutralization of 13 isolates sensitive to PGZL1 gVmDmJ chosen from the 130-virus panel. Source data for **b**–**g** are provided as a Source Data file.

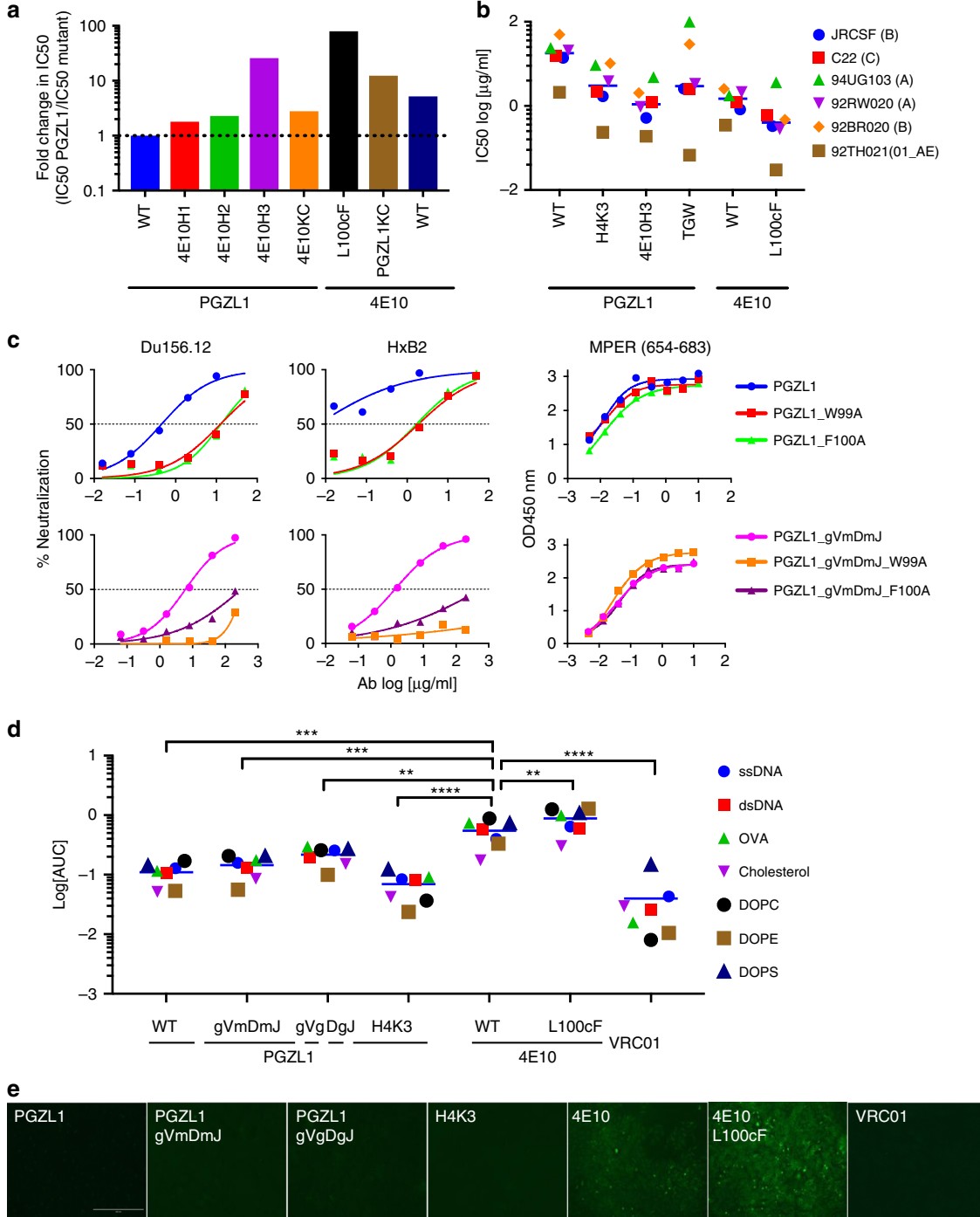

**Fig. 3** Dominant role of CDRH3 in PGZL1 HIV-1 neutralization by D-gene-encoded residues. **a** Fold decrease in neutralization (IC$_{50}$) of isolate 92TH021 relative to wild-type PGZL1 by CDR grafts and LC substitution from 4E10 (left), and by 4E10 substitutions L$_{100c}$F and PGZL1 LC (right). **b** Neutralization (log IC$_{50}$) of a six-virus panel by PGZL1 and 4E10 variant antibodies. **c** Effect of Ala substitutions in $D_H$-encoded residues W99 and F100 on the ability of PGZL1 mature (top panels) and inferred germline antibodies (bottom panels) to neutralize Du156.12 and HxB2 (left panels), as well as to bind MPER peptide in an ELISA (right panels). **d** Antibody polyreactivity in an ELISA as a function of area under the curve (AUC) of PGZL1 and 4E10 variant antibodies against nonspecific antigens. VRC01 is a negative control. Two-way ANOVA multiple comparisons was used to compare the difference between groups ($n = 7$, **$p = 0.0062$, ***$p = 0.0001$, ****$p < 0.0001$). **e** Immunofluorescence staining of HEp-2 cells. Antibodies were tested at 50 μg/ml using 4E10 and VRC01 as positive and negative controls, respectively; images are at ×200 magnification and the scale bar is 400 μm. Source data for **b**–**d** are provided as a Source Data file.

the most potent 4E10 mutant of those we tested or could find in the literature.

As PGZL1 and 4E10 share the $D_H3$-10*01 germline gene (Supplementary Fig. 4a), we asked whether Ala mutants of

$D_H3$-10*01-encoded W$_{99}$ and F$_{100}$ affect the activity of PGZL1 and PGZL1 gVmDmJ. Mutations W$_{99}$A and F$_{100}$A decreased neutralization of Du156.12 and HxB2 by these antibodies more than tenfold (Fig. 3c). Notably, W$_{99}$A and F$_{100}$A did not affect

binding of either antibody to MPER peptide in enzyme-linked immunosorbent assay (ELISA) (Fig. 3c), suggesting their activity may relate more to MPER recognition on virions. Hence, $D_H3$-$10*01$-encoded $W_{99}$ and $F_{100}$ are crucial for neutralization by PGZL1.

BnAb 4E10 exhibits polyreactivity[8], which is of interest, as 4E10 knock-in mice show B-cell tolerance control[9,24]. By ELISA, we verified that 4E10 was polyreactive towards single-stranded DNA, double-stranded DNA (dsDNA), ovalbumin (Ova), and lipids, cholesterol, 1,2-dioleoyl-sn-glycero-3-phospho-choline (DOPC), 1,2-dioleoyl-sn-glycero-3-phosphoethanola-mine (DOPE), and 1,2-dioleoyl-sn-glycero-3-phospho-L-serine (DOPS), whereas a negative control bnAb VRC01 showed no such binding (Fig. 3d). Of note, 4E10 $L_{100c}F$ showed higher binding than 4E10 to the above panel. PGZL1, gVmDmJ, gVgDgJ, and H4K3, all showed less nonspecific binding than 4E10 to the same antigens. Similarly, HEp-2 cells were stained by 4E10, whereas PGZL1, gVmDmJ, gVgDgJ, and H4K3, all showed less staining (Fig. 3e).

**PGZL1 structure closely resembles 4E10**. To gain insight into PGZL1 three-dimensional (3D) structure and mode of binding, we determined crystal structures of unbound (1.4 Å resolution) and MPER$_{671-683}$-bound PGZL1 (3.65 Å) Fabs (Fig. 4a, b and Supplementary Table 7). The superposition of Cα atoms of the two PGZL1 variable domains yielded a root means square deviation (r.m.s.d.) of 0.6 Å (Fig. 4a). Except for CDRH3 where residues are displaced by up to ~ 4.8 Å (i.e., $W_{99}$), bound and unbound Fabs show only minor differences in CDR loops. PGZL1 resembles the 4E10 structure (PDB 2FX7 [https://www.rcsb.org/structure/2fx7])[13]) (Cα r.m.s.d. of ~ 0.4 Å; Fig. 4b). PGZL1 CDRH3 has a well-defined density, even in the absence of pep-tide, compared with 4E10's three-residue longer CDRH3, where the residues $G_{99}WGW_{100b}$ at its tip were less defined. $W_{99}$ in PGZL1 is ~ 7.8 Å apart from the equivalent $W_{100b}$ in 4E10 (Supplementary Fig. 1a), neither of which contact MPER$_{671-683}$ (Fig. 4b–d; PDB 2FX7 [https://www.rcsb.org/structure/2fx7]). 4E10 $W_{100b}$ may contact the TM region of gp41[28], but the shorter CDRH3 makes TM contact unlikely with PGZL1. However, $F_{100}$ of PGZL1 CDRH3 (Fig. 4c), which corresponds to $L_{100c}$ in 4E10, may re-orient to make aromatic interactions with MPER $W_{680}$ and $Y_{681}$ in the viral membrane.

PGZL1 binds to MPER$_{671-683}$ using similar contact residues as 4E10 in CDRs H1, H2, H3, and L3 (Fig. 4c, d). In both complex structures, MPER$_{671-683}$ is helical with $W_{672}$-$D_{674}$ in a capping $3_{10}$ helix, and germline-encoded $Y_{91}$ (LC) and $F_{102}$, $W_{47}$ (HC) form an aromatic patch with peptide $F_{673}$ and $W_{672}$. Another aromatic patch is seen near MPER $W_{680}$ and $Y_{681}$, but the paratope residues differ between PGZL1 and 4E10. In the PGZL1 structure, $F_{100}$ ($L_{100}$ in 4E10) is close to these MPER aromatics. In the 4E10 structure, germline-encoded $Y_{32}$ of CDRH1 ($L_{32}$ in PGZL1) is also involved in this patch (Fig. 4c, d). The greater potency observed with 4E10 $L_{100c}F$ may be due to improved interaction of its CDRH3 with MPER and membrane by adding $F_{100c}$ to this hydrophobic patch. Some hydrogen bonds between 4E10 and MPER residues are also observed in PGZL1, but the lower resolution of the PGZL1-MPER$_{671-683}$ structure (3.65 Å) pre-cludes a precise count.

**Unbound H4K3 is precisely preconfigured to bind MPER**. Crystal structures of H4K3 unliganded and with MPER$_{671-683}$ bound were determined at 1.45 Å and 1.98 Å resolution, respec-tively. Strikingly, their variable regions, including CDR loops, are in nearly identical conformations (Cα r.m.s.d. ~ 0.2 Å), which are both similar to the MPER-bound PGZL1 (Cα r.m.s.d. ~ 0.4 Å),

suggesting H4K3 is already preconfigured for MPER binding (Fig. 4e). The shorter and more rigid CDRH3 of H4K3 may explain its lower polyreactivity compared with 4E10 whose CDRH3 is more flexible and hydrophobic.

PGZL1 and H4K3 structures closely resemble VRC42.01 structure (PDB 6MTO [http://www.rcsb.org/structure/6MTO]; Cα r.m.s.d. of the variable domains ~ 0.6 Å). However, a slight difference in MPER binding is observed at the N-terminal region of the epitope (residues 671–675; Supplementary Fig. 5a). All MPER epitope residues are in nearly identical positions when bound to PGZL1, H4K3, and 4E10 (Supplementary Fig. 5c, d). In contrast, the N-terminal region is shifted slightly away from the combining site in VRC42.01 (1.1 Å difference between the Cα of MPER $F_{673}$ in PGZL1 and VRC42.01, and as much as 1.8 Å in the aromatic ring position; Supplementary Fig. 5b). It is noteworthy that the MPER peptide used in our crystal structures differs at position 677 from that in the VRC42.01 complex: asparagine vs. lysine, respectively (Supplementary Fig. 5b). This residue is located at the periphery of the combining site where the MPER position is similar in both structures. Thus, the different positioning of the N-terminal region of the MPER might be due to difference in residues of the J gene region between PGZL1 and VRC42.01 (Supplementary Fig. 1a and Supplementary Table 2). Phenylalanine at the position 100 g in the CDRH3 loops of PGZL1, H4K3, and 4E10 form an aromatic cluster with CDRL3 $Y_{91}$ and MPER $F_{673}$. In VRC42.01, this aromatic cluster is less tightly packed due to a methionine at CDRH3 position 100 g (Supplementary Fig. 5b-d).

**PGZL1 neutralization resistance mutations and MPER Ala scan**. To identify resistance mutations to PGZL1, we first per-formed long-read NGS analyses on *env* rescued from con-temporaneous PG13 donor plasma. Sequences were determined for 58 clade B Envs. One small sublineage and a larger, more diverse clade were predicted to be CXCR4 tropic and CCR5 tropic, respectively (Supplementary Fig. 6a). Notably, MPER polymorphisms $D_{674}S$, $D_{674}T$, and $D_{674}E$ are predicted to contact PGZL1 and are distinct from other reported 4E10 resistance mutations, $F_{673}L$ or $W_{680}G/R$[29,30] (Supplementary Fig. 6a). As viruses pseudotyped with PG13 Envs lacked infectivity, we tested the MPER polymorphisms in the context of primary isolate COT6, while also testing PGZL1 and 4E10 against a COT6 MPER Ala mutant virus panel. Indeed, $D_{674}S$, $D_{674}E$, and $D_{674}T$, all made COT6 more resistant to neutralization by PGZL1, H4K3, and 10E8 (Table 1). Mutant $D_{674}A$ was resistant to PGZL1 and H4K3, but less so with 4E10. Some PG13 Envs have Gly at MPER position 662, which is present in 1.2% (vs. Glu in 71.7%) of Envs in the LANL database (Supplementary Fig. 6c). However, $A_{662}G$ resulted in COT6 being tenfold more sensitive to PGZL1 and 4E10 (Supplementary Fig. 6b). All COT6 mutants were similarly sensitive to control antibody VRC01. Overall, we infer that some viruses in the PG13 donor developed polymorphisms in $D_{674}$ to resist PGZL1 antibodies.

COT6 Ala mutants $W_{672}A$, $F_{673}A$, and $W_{680}A$ were resistant to PGZL1, HK43, and 4E10 (Supplementary Fig. 6c), as well as to 10E8[31,32]. However, MPER Ala mutations rarely occur naturally (Supplementary Fig. 6c). Fold changes in IC$_{50}$ differed among the 4E10-like bnAbs against mutants $S_{671}A$, $D_{674}A$, $I_{675}A$, and $L_{679}A$ (Supplementary Fig. 6b). Thus, IC$_{50}$s of H4K3 and 4E10 against $S_{671}A$ and $L_{679}A$ differed by up to 100-fold, despite a cross-clade correlation in their IC$_{50}$s (Supplementary Fig. 3d). In H4K3 and 4E10-bound forms, $S_{671}A$ has a similar structural environment, so this mutation might impact MPER folding or accessibility. However, neutralization differences with COT6 $L_{679}A$ seem more likely to be due to HC position 54, which is occupied by a valine, phenylalanine, and leucine in H4K3, PGZL1, and 4E10,

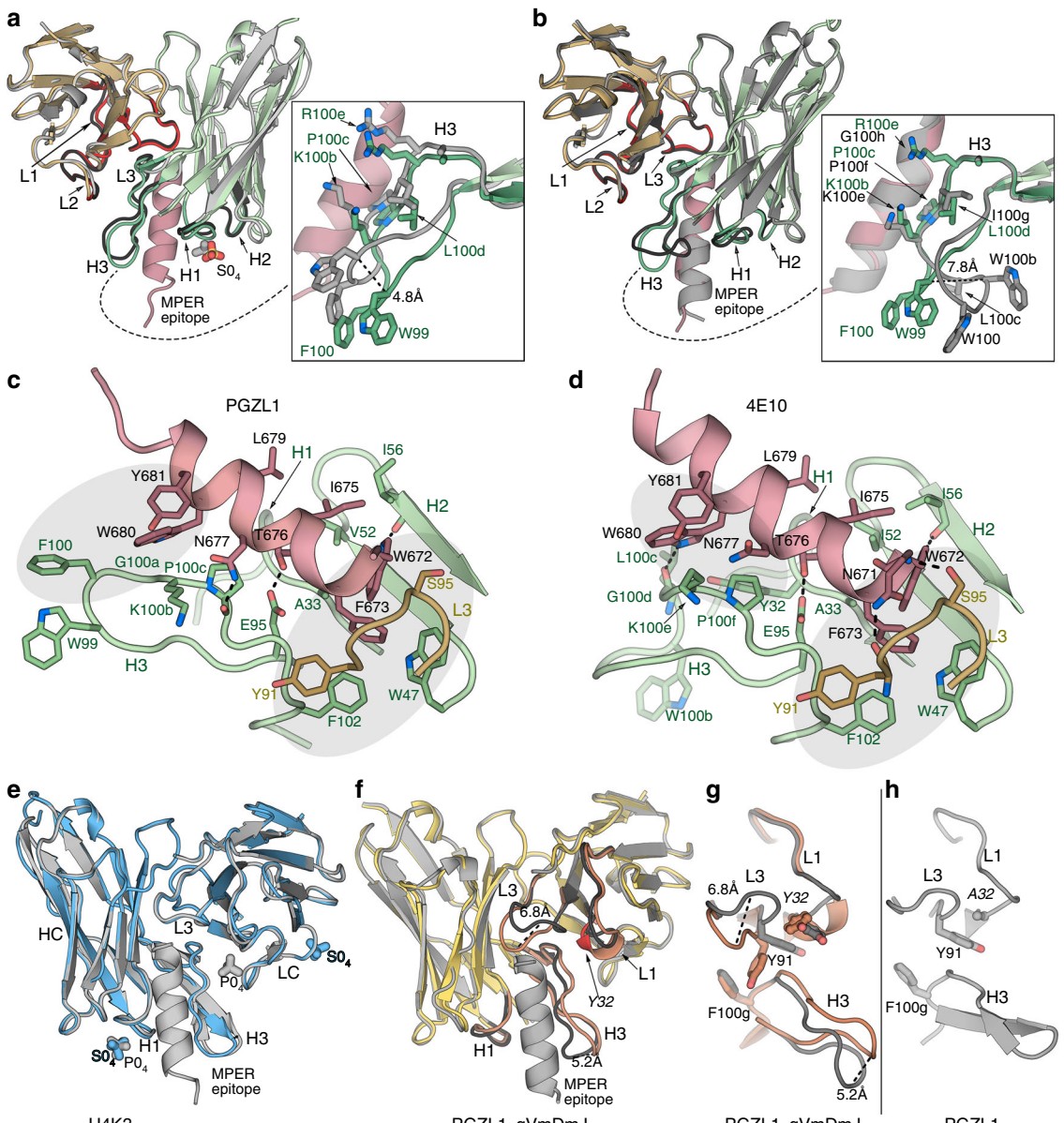

**Fig. 4** PGZL1 variant crystal structures and comparison with 4E10. **a** Superposition of the crystal structures of the mature PGZL1 variable domain (from the Fab) bound to MPER$_{671-683}$ (wheat, LC; green, HC; pink, MPER) and unbound PGZL1 (gray). CDRs of the bound structure are shown in red (LC) and green (HC), and CDRs of the unbound structure are shown in black. Inset: superposition of free and bound CDRH3 with residues near the MPER shown as sticks. **b** Superposition of PGZL1–MPER$_{671-683}$ and 4E10-MPER$_{671-683}$ (gray; PDB 2FX7 [https://www.rcsb.org/structure/2fx7][13]). Coloring and inset as in **a**. **c** PGZL1–MPER$_{671-683}$ combining site (wheat, LC; green, HC; pink, MPER; interacting residues - sticks). Shaded regions highlight aromatic clusters. **d** 4E10-MPER$_{671-683}$ combining site, colored as in **c**. **e** Superposition of unbound (blue) and MPER$_{671-683}$-bound (gray) H4K3. Ions are shown as sticks. **f** Superposition of unbound (yellow, CDR loops, brown) and MPER$_{671-683}$-bound (gray) PGZL1 gVmDmJ. **g** CDR loops in bound (gray) and unbound (brown) PGZL1 gVmDmJ with residues that influence loop conformations shown as sticks. **h** Same region as in **g** for the mature PGZL1 structure.

**Table 1 Effect of MPER mutation on the neutralization of HIV COT6 by PGZL1 and control antibodies.**

| Virus | MPER epitope | COT6 mutants | Fold increase in IC$_{50}$ | | | | |
|---|---|---|---|---|---|---|---|
| | | | **PGZL1** | **H4K3** | **4E10** | **10E8** | **VRC01** |
| COT6 | SWFDITKWLW | WT | 1 | 1 | 1 | 1 | 1 |
| PG13 Cons | NWFDITNWLW | | NA | NA | NA | NA | NA |
| PG13 isolate 1 | NWFSITNWLW | D674S | 7.7 | 6.29 | 1.65 | NA | NA |
| PG13 isolate 2 | NWFEITNWLW | D674E | 26 | 12 | 10 | 17.5 | 1.2 |
| PG13 isolate 3 | NWFTITNWLW | D674T | 120 | 38 | 25 | 36.0 | 1.0 |

*NA*, not attempted

respectively; structural analysis of this region reveals more tightly packed interactions between the MPER and 4E10 vs. PGZL1 or H4K3. Of note, MPER mutations L663A, D664A, S665A, W666A, K667A, L669A, W670A, K677A, and W678A enhanced HIV neutralization by the three MPER bnAbs, as also reported previously for 4E10[32]. The neutralization enhancement effect with these mutants is not fully understood but might be explained by changes in Env conformation that increase accessibility and/or susceptibility to functional inactivation by the MPER bnAbs[31].

**PGZL1 inferred germline structure**. The $MPER_{671-683}$-bound (2.47 Å) and -unbound (2.6 Å) structures of inferred germline PGZL1 gVmDmJ notably differ in the conformation of their CDRs L1, L3, and H3 (Fig. 4f). When superimposed, residues in CDRL3 in the unbound structure are shifted up to 6.8 Å from those in the bound complex. This shift of CDRL3 in the bound structure may arise from MPER binding, which cause a rearrangement in the aromatic interaction between $Y_{32}$ (CDRL1) and $Y_{91}$ (CDRL3) (Fig. 4g). We note that $Y_{32}$ has affinity matured in PGZL1 to alanine, which does not affect $Y_{91}$'s orientation; thus, the CDRL3 and L1 conformations are similar in both unbound and bound PGZL1, and resemble the bound form of the inferred germline PGZL1 Fab (Fig. 4h).

**Structural and functional evidence of PGZL1 membrane binding**. To elucidate potential membrane interactions of PGZL1, we solved crystal structures of PGZL1-$MPER_{671-683}$ and unliganded H4K3 in the presence of a short acyl tail phosphatidic acid (06:0 PA) at 3.42 Å and 3.11 Å resolution, respectively. We observed lipid-binding sites in both antibody structures (Fig. 5a, b). The first lipid site in PGZL1 and H4K3 is in a similar location in 4E10[6], proximal to CDRH1 $S_{28}$, $F_{29}$, and $S_{30}$ (Fig. 5d and Supplementary Fig. 7a). Next to this site in H4K3, a second PA lipid interacts with $D_{72}$, $R_{73}$, and $S_{74}$ in the HC framework region FRH3. Of note, in H4K3, these 2 lipids are part of a ~ 33 Å lipid vesicle formed at the interface of 12-symmetry-related Fabs that each contain the 2 lipid sites (Fig. 5c), as also observed in the 4E10 structure[6].

$PO_4$ and $SO_4$ ions were both bound at the CDRH1 site of the H4K3 structures obtained in lipid-free buffer (Fig. 4e). Another $PO_4$ ion interacts with $R_{100b}$ and $R_{100e}$ of CDRH3 and $Y_{91}$ of CDRL3 in the H4K3-$MPER_{671-683}$ structure (Figs. 4e and 5e, and Supplementary Fig. 7b). As this anion-binding site is absent in peptide-free H4K3, despite the crystallization buffer containing $SO_4$, this site may be induced by MPER binding. This site is also H4K3 specific, as it is absent in peptide-bound PGZL1, where LC SHM-residue $D_{50}$ replaces $G_{50}$ (Fig. 6a, b) and residue $100_b$ is lysine. Another $SO_4$ ion was observed in unbound H4K3 proximal to FRL3 and interacts with $P_{59}$, $G_{60}$, and $R_{61}$ (Fig. 5f and Supplementary Fig. 7c). Thus, these two anion-binding sites may also represent phospholipid-binding sites.

To determine whether the lipid-binding sites observed in our structures are biologically relevant, we mutated the H4K3 lipid-binding sites. Both CDRH1 and FRH3 regions form plateau-like structures with main-chain nitrogens making contacts with the lipid head groups (Fig. 5a, b, d, f). We thus mutated the lipid-binding regions $S_{28}F_{29}S_{30}$ of CDRH1 and $D_{72}R_{73}S_{74}$ of FRH3 to $E_{28}P_{29}E_{30}$ and $D_{72}P_{73}E_{74}$, respectively, with the proline mutation chosen to perturb the lipid interaction with the main chain and the glutamate mutation to introduce negative charges that are repulsive to the viral membrane. We also mutated residue $G_{50}$ of the H4K3 LC to $D_{50}$, to disrupt the CDRH3 $R_{100b}$-$R_{100e}$ site. In summary, we created the following three mutants: (1) H4K3_SFS28EPE that includes $E_{28}P_{29}E_{30}$ in the HC and $D_{50}$ in the LC; (2) H4K3_RS73PE that includes $P_{73}E_{74}$ in the HC and

$D_{50}$ in the LC; and (3) H4K3_5M that includes all five mutations in the HC combined with $D_{50}$ in the LC. ELISA and BLI experiments show that the three mutants retain similar nM binding affinity to the MPER peptide as H4K3 (Fig. 6c, e). However, the neutralization potency ($IC_{50}$) against HxB2 and Du156.12 is reduced with all three mutants by 7–84-fold, with the H4K3_SFS28EPE and H4K3_5M exhibiting higher losses in potency (Fig. 6d, e). Hence, mutation of the lipid-binding sites influences binding to membrane-embedded Env on the virus but not binding to MPER out of the membrane context.

Analysis of the surface potential of PGZL1 variants and 4E10 show that the lipid head groups and anions bind to electropositive clefts on the antibody surface (Fig. 6f–i). Germline PGZL1 gVmDmJ and mature PGZL1 have less basic patches compared with H4K3 and 4E10. Thus, their lower neutralization potency may correlate with reduced electrostatic interaction with specific lipids in the viral membrane.

Considering the position and orientation of lipids, anion-binding sites (Fig. 5b–g), and MPER orientation (Fig. 4e), we generated a molecular model of H4K3 binding to the MPER-viral membrane using CHARMM force field and molecular dynamics (MD) simulation, in a similar way to the published 4E10 and 10E8 models[6,33,34]. The lipid head groups of the membrane outer leaflet were placed in a plane that roughly includes $K_{683}$ of gp41 as well as the two lipid head groups and anion sites observed in our crystal structures (Fig. 5g). Our model suggests that H4K3 contacts the membrane-MPER epitope in a similar manner to 4E10[6] with the MPER helix tilted 67°–73° from the bilayer surface. CDRH1, CDRH3 tip, FRH3 (residues 73–76), N-terminal HC residue $K_1$, and CDRL2 and FRL3 (residues 56–61 and $R_{77}$) are proximal to and interact with the lipid heads. A hydrophobicity plot showed similar membrane-transfer propensity between PGZL1, H4K3, VRC42.01, and 4E10 in both HCs and LCs, except that VRC42.01 and 4E10 CDRH3 showed higher hydrophobicity (Supplementary Fig. 1b).

**Cryo-EM structure of Env-PGT151-PGZL1 complex**. To explore the topology of the PGZL1-Env interaction, we obtained a cryo-EM reconstruction at 8.9 Å resolution of a full-length Env of isolate AMC011[35] in complex with PGZL1 and PGT151. Despite notable structural heterogeneity between the well-resolved ectodomain vs. detergent–lipid micelle containing the TM and cytoplasmic tail, we classified a subset of particles and reconstructed a 3D map with one PGZL1 Fab bound to one MPER of the AMC011 trimer (Fig. 5h and Supplementary Fig. 6). Two PGT151 Fabs were also observed, whereas the gp41 TM region cannot be distinguished from the detergent–lipid micelle. Fitting of the PGZL1 lipid-bound Fab crystal structure (Fig. 5g) into the cryo-EM map confirms the antibody orientation relative to the bilayer and suggests both HC and LC contact the membrane (Fig. 5h). The angle of approach revealed in the MD simulation (Fig. 5g) also concurs with the cryo-EM docking model.

## Discussion

HIV bnAbs that share epitope and paratope features have been of increasing interest for vaccine design[2]. The bnAbs 4E10, 10E8, and DH511 show near pan-neutralization of HIV-1 by targeting nearly identical α-helical epitopes in the MPER, but these antibodies have uncommon features. We described PGZL1 and its lineage variant H4K3 and germline revertants that have some common features that may aid vaccine design, especially given the homology to 4E10 and the recently described VRC42 antibody. PGZL1 antibodies have a 15-residue CDRH3 that is close to the average in humans of ~ 13.2[7]. Similar to all MPER bnAbs, CDRH3 aromatics were crucial for HIV neutralization, but the

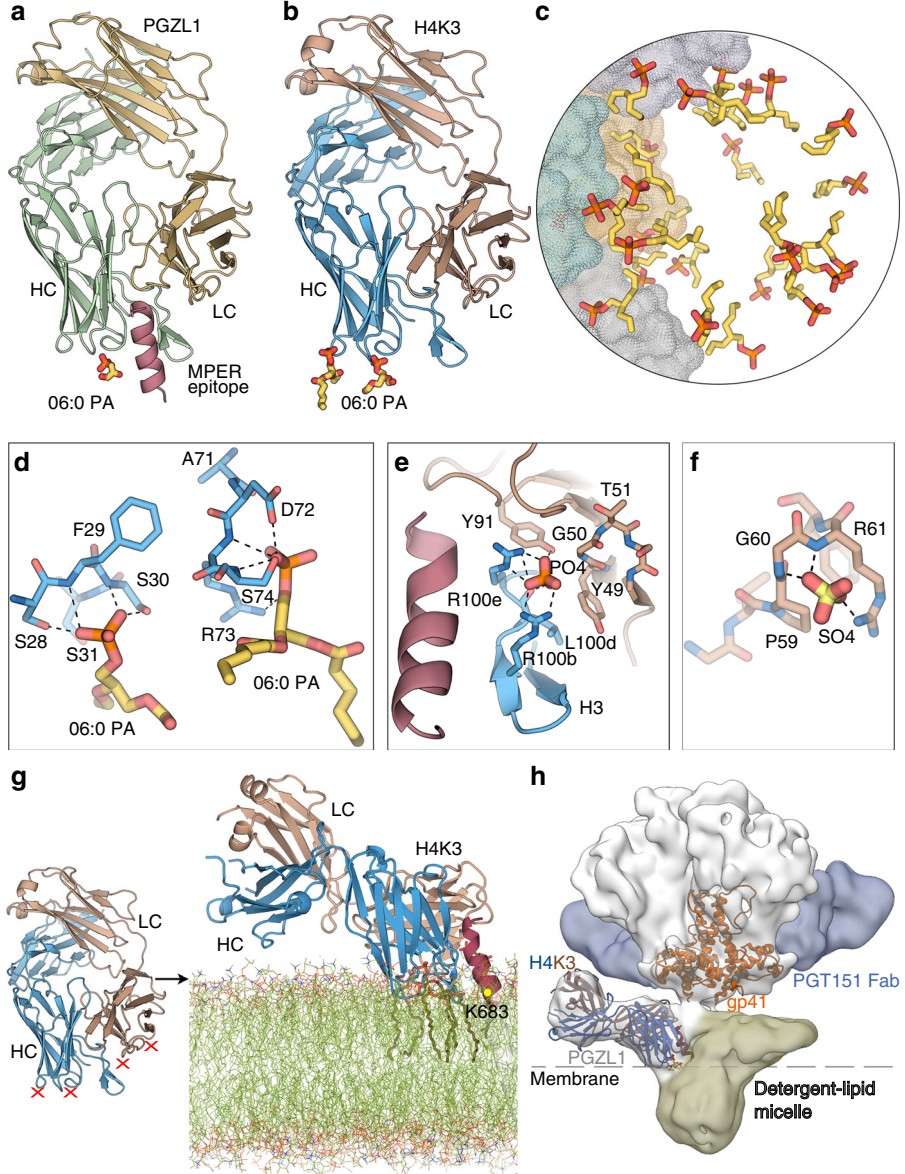

**Fig. 5** Lipid binding and angle of approach of PGZL1 variants to the viral membrane. **a** Cartoon of the PGZL1-MPER$_{671-683}$ complex structure (wheat, LC; green, HC; pink, MPER) crystallized with 06:0 PA (sticks). **b** Cartoon of H4K3 (brown, LC; blue, HC) crystallized with 06:0 PA (sticks). **c** Stick rendering of 06:0 PA fragments forming a lipid vesicle at the interface of 12 crystallographic and non-crystallographic-related H4K3 Fabs. The four Fabs in the asymmetric unit are shown as gray, green, yellow, and blue color surfaces. **d** Stick rendering of observed lipid-binding sites in H4K3. **e** Phosphate-binding site near to H4K3 CDRH3 when MPER$_{671-683}$ (pink) is bound. Colors as in **b**. **f** Sulfate-binding site in FRL3 of H4K3. **g** Model of H4K3 binding to the MPER (red)-viral membrane (green) epitope. The model at the right side of the arrow was built based on the regions where experimental lipids and anions (red X) bind on H4K3 (left side of the arrow); cognate lipids are shown as sticks inside the modeled membrane. The position of the MPER K683 residue is indicated with a yellow dot. **h** Cryo-EM reconstruction of full-length AMC011-PGT151-PGZL1 complex at 8.9 Å with H4K3 (blue/brown ribbons) fitted into the Env density at the base of the gp41 stem. The MPER is shown as a red ribbon and lipid head groups as sticks. The detergent-lipid micelle is shown in olive and PGT151 density in blue. Dashed lines show the approximate location where the outer surface of the membrane would be on the virus or infected cells.

observed pre-configuration of H4K3 to bind the MPER and lipid heads on the membrane may explain its faster on-rate compared with PGZL1, where reorientation of the CDRH3 is needed. An induced lipid site at residues R$_{100e}$/R$_{100b}$ of H4K3 may also have improved its potency while reducing polyreactivity. By comparison, longer CDRH3s on 4E10 and VRC42 variant VRC42.N1 may interact more with the TM domain in the upper membrane core[28]. Using X-ray crystallography and cryo-EM, we determined the orientation, angle of approach, and composition of the PGZL1-Env interaction. These valuable data can inform the

design of HIV vaccines and therapies, involving a vulnerable site that has long precluded a complete description due to ambiguity of the membrane interaction.

PGZL1, 4E10, and VRC42 are derived from similar V/D genes but different J genes. Whether other shared CDRH3 residues arose from SHM or N1/N2 additions is unknown but should be considered, as they may affect antibody activity. Vaccine design must also consider how to elicit bnAbs with few SHM. Fortunately, the SHM-reverted PGZL1 gVmDmJ antibody neutralized up to 28% of viruses tested, albeit mostly with limited

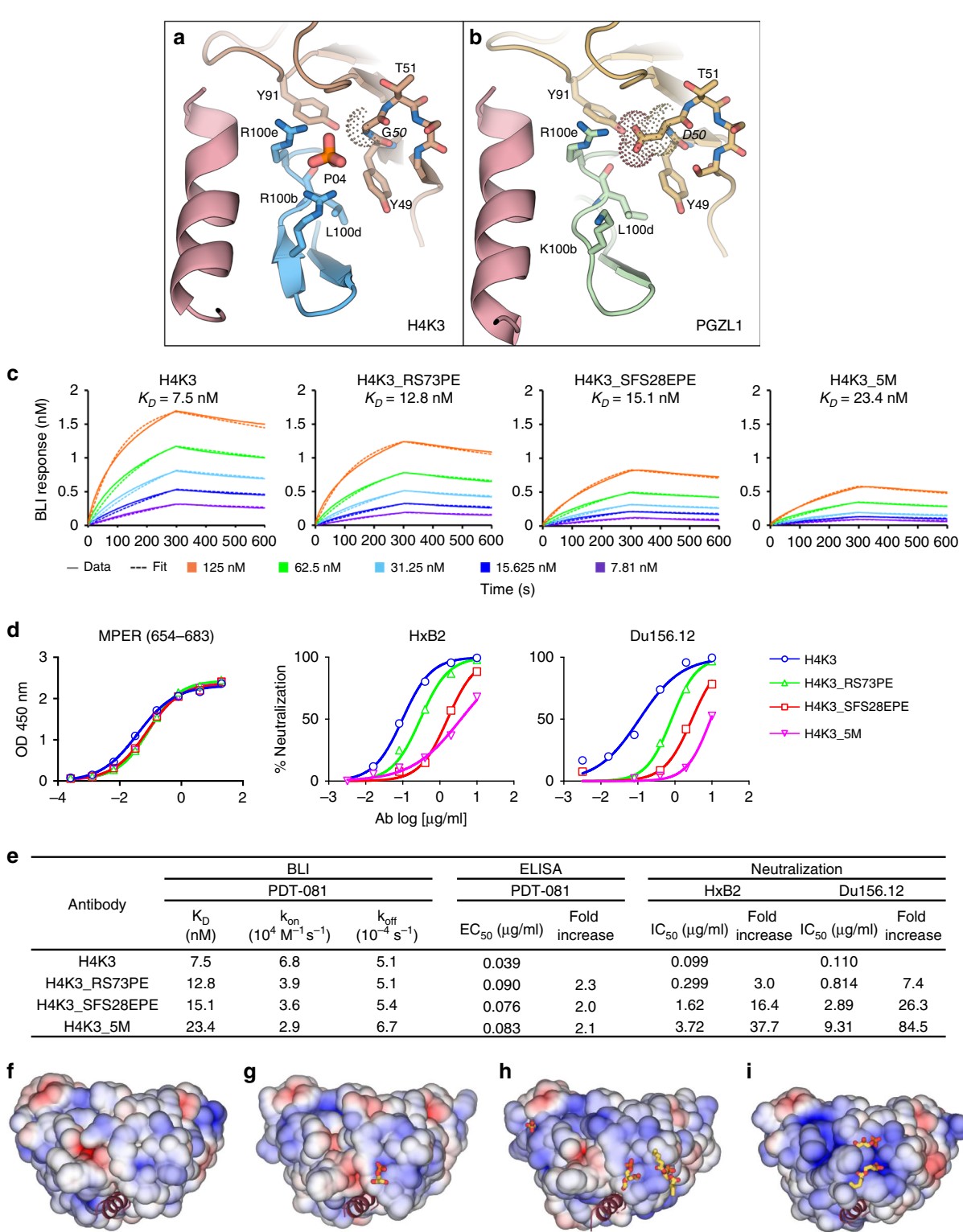

**Fig. 6** MPER-induced $PO_4$-binding site of H4K3 and PGZL1 lipid site mutant characterization and electrostatics. **a**, **b** Stick rendering of the $PO_4$-binding site in (**a**) H4K3 and (**b**) PGZL1. The side chain at LC position 50 in each antibody is surrounded by dots. Select HC residues of the two antibodies are shown in blue (**a**) and green (**b**), and the LC is shown in brown. MPER is shown as a pink ribbon. **c** Binding of H4K3 lipid-binding site mutants to immobilized MPER peptide by BLI. **d** Binding of H4K3 lipid-binding site mutants to MPER peptide by ELISA and neutralization (log $IC_{50}$) of HxB2 and Du156.12. **e** BLI, ELISA binding to MPER peptide, and neutralization statistics. **f**–**i** Surface rendering, along the MPER helical axis (red ribbon), of the solvent accessible electrostatic potential contoured at ± 5 kT/e for (**f**) PGZL1 gVmDmJ, (**g**) PGZL1, (**h**) H4K3, and (**i**) 4E10. Observed lipid fragments and anions are shown as sticks. Source data for **h** and **i** are provided as a Source Data file.

potency; many somatic mutations in 4E10 and 10E8 are also non-essential[18,19]. $V_H1$-69 is also highly mutable due to several SHM hotspots[36]. $V_H1$-69 has been associated with autoantibodies of leukemia[36], HCV-associated mixed cryoglobulinemia[37], as well as bnAbs highly specific to the influenza hemagglutinin stem[38] and HCV E2[39]. Thus, any vaccine strategy to elicit $V_H1$-69 MPER bnAbs must be MPER specific and avoid or limit activation of off-target B cells.

We found a few Envs that were recognized and neutralized by germline revertants of PGZL1 and 4E10. Such Envs, or perhaps the T/F Env of the VRC42 donor[5], might be useful for activating 4E10-like B-cell precursors whose CDRH3s can engage membrane-bound Env. These B cells could be boosted with a heterologous Env to drive the maturation process required for achieving neutralization breadth and potency (Pathway 1). Alternatively, $V_H1$-69 restricted B cells that lack crucial lipid-interacting residues might be activated, perhaps using specific Envs, scaffolds, peptides, or anti-idiotype antibodies[3,40–42]. Lipid-interacting residues would have to be provided by the maturation process and the B cells boosted perhaps by a membrane-anchored MPER (Pathway 2). Contrarily, 10E8 and DH511 do not bind Env as germline revertants; thus, specific strategies may be needed to activate such lineages. As soluble Env differs somewhat from its membrane counterpart in glycosylation and conformation[43–45], development of membrane-embedded Env vaccines or suitable soluble alternatives seems justified[25].

H4K3 lacks the potency of bnAbs currently being developed for therapy, such as 10E8[10]. However, its exceptional breadth warrants efforts to engineer H4K3 to be more potent without enhancing polyreactivity, as was successfully accomplished recently with 10E8[46,47].

MPER bnAbs have been associated with IgG3-prone B-cell subsets[4]. PGZL1 shows that IgG1 MPER B cells can also be elicited in humans, which we speculate arose from a precursor with minimal polyreactivity that bound directly to Env. Non-IgG3 MPER nAbs have notably also been observed in donor plasma[48] and variants of VRC42 are both IgG1s and IgG3s. Whether such differences in isotype, polyreactivity, or autoreactivity of 4E10-like bnAbs translate into useful correlates for eliciting MPER bnAbs requires future immunization studies.

## Methods

**Ethics statement, study participant, and samples**. Donor PG13 was from the IAVI-sponsored Protocol G cohort in South Africa[20]. Blood samples were collected with written, informed consent, and the study was reviewed and approved by the relevant Ethics and Research Committees.

**Isolation of PGZL1 monoclonal antibodies**. Fluorescent-labeled antibodies CD3-Alexa Fluor 700 (BD 557943), CD8-Alexa Fluor 700 (BD 561453), CD14-PE-Cy7 (BD 561385), CD16-PE-Cy7 (BD 560716), CD19 PerCP*Cy5.5 (BD 561295), CD20 PerCP*Cy5.5 (BD 350955), IgG-PE-Cy5 (BD 551497), IgM-BV605 (BD 562977), and IgD-BV605 (BD 563313), which target cell surface markers, were used as 1:200-fold dilution in staining. Biotin-labeled MPER peptide PDT-081 ($E_{654}$KNEQELLELDKWASLWNWFDITNWLWYIK$_{683}$-biotin) was purchased from GenScript and coupled separately to streptavidin-BV421 (BD 562426) and streptavidin-APC (allophycocyanin, BD 555335). PBMCs were stained using the LIVE/DEAD Fixable Near-IR Dead Cell Kit (Life Technologies, L34957) for 30 min on ice. Cells were then labeled with antibodies cocktail along with MPER probes for 1 h in Brilliant Staining buffer (BD 563794) on ice. Cell population CD19+/CD20+, CD3−/CD8−, CD14−/CD16−, IgG+, IgD-/IgM− MPER double positive were sorted using BD FACSAria III sorter into individual wells of a 96-well plate containing lysis buffer and plates were immediately sealed and stored at −80 °C. The single B-cell sorting strategy is shown in the Supplementary Fig. 9.

**Antibody sequence amplification, analysis, and cloning**. The first-strand complementary DNA from B cells was synthesized using Superscript III Reverse Transcriptase (Life Technologies) and random hexamers (Gene Link). Nested PCR amplification of HC and LC variable regions was performed using Multiplex PCR Kit (Qiagen). Amplified HC and LC variable regions were sequenced and then analyzed using IMGT online tools. Antibodies of interest were cloned into

expression vectors by re-amplification of the variable regions using the same primers but modified to introduce homology to the vector[49]. Primers used are reported in the Supplementary Table 8.

**PG13 antibody repertoire sequencing and analysis**. The 5′-RACE (rapid amplification of cDNA ends) PCR protocol used for unbiased human B-cell repertoire analysis has been previously described[22,50]. Briefly, total RNA was extracted from fivemillion PBMCs into 30 μl of water with RNeasy Mini Kit (Qiagen). 5′-RACE was performed with SMARTer RACE cDNA Amplification Kit (Clontech). The immunoglobulin PCRs were set up with Platinum Taq High-Fidelity DNA Polymerase (Life Technologies) in a total volume of 50 μl, with 5 μl of cDNA as template, 1 μl of 5′-RACE primer, and 1 μl of 10 μM reverse primer. The 5′-RACE primer contained a PGM/S5 P1 adaptor, whereas the reverse primer contained a PGM/S5 A adaptor. A total of 25 PCR cycles were performed and the expected PCR products ( ~ 600 bp) were gel purified (Qiagen). NGS was performed on the Ion S5 system as previously described[21]. Briefly, heavy (H), kappa (κ), and lambda (λ) chain libraries were quantified using Qubit® 2.0 Fluorometer with Qubit® dsDNA HS Assay Kit, and were mixed at a ratio of 2:1:1 before antibody libraries from three time points were further mixed at a ratio of 1:1:1. Ion Xpress$^{TM}$ barcodes (Life Technologies), #1–#3, were used to tag antibody libraries to differentiate the three time points. Template preparation and (Ion 520) chip loading were performed on Ion Chef using the Ion 520/530 Ext Kit, followed by sequencing on the Ion GeneStudio S5 platform with default settings. Raw data were processed without the 3′-end trimming in base calling to extend the read length. An improved version of the Antibodyomics pipeline[21] was used to process, annotate, and analyze the sequencing data of PG13 antibody repertoires. After pipeline processing, a bioinformatics filter was applied to remove erroneous sequences that may contain swapped gene segments due to PCR errors. Specifically, a full-length variable region sequence would be removed if the V-gene alignment was <250 bp. The results for pipeline processing are summarized in Supplementary Table 3. The pipeline-processed antibody chain sequences were subjected to two-dimensional (2D) divergence/identity analysis and CDR3-based lineage analysis, with putative somatic variants determined at CDR3 identity cutoffs of 80% and 95% (Fig. 1d).

**Expression and purification of PGZL1 IgG and Fab variants**. PGZL1 HCs and LCs were cloned using Gibson Assembly Enzyme mix (NEB) into expression vectors with the appropriate IgG1, Igκ, or Igλ constant domains[49]. Antibodies were expressed in FreeStyle 293F cells (Life Technologies Cat#R79007). Briefly, ~ 750 μg DNA (500 μg HC and 250 μg LC plasmid) were added to 25 ml Opti-MEM (Life Technologies, 31985-070), which was mixed with Opti-MEM containing 2250 μg polyethylene imine MAX (molecular weight 40,000 kDa; Polyscience, 24765-1). After incubation for 20 min at room temperature (RT), the transfection mix was added to 1 L cells at a density of ~ $1.2 \times 10^6$ cells/ml in FreeStyle293 Expression Medium (Life Technologies, 12338018). The cells were incubated at 37 °C and 8% $CO_2$ for 6 days. After collecting the cells, the supernatant, containing IgG or Fab, was filtered and loaded into a protein A beads column (Thermo Scientific) or HiTrap KappaSelect column (GE Healthcare Life Sciences, 17545812). The column was washed with phosphate-buffered saline (PBS) and eluted with 0.2 M citric acid pH 3.0 or 0.1 M glycine pH 2.7. The fractions were concentrated and the buffer was changed to 20 mM sodium acetate pH 5.5. The Fab was loaded into a Mono S column and was eluted with a 0–60% linear gradient of 1 M sodium chloride and20 mM sodium acetate pH 5.5 buffer. The Fabs were concentrated and stored in 20 mM sodium acetate pH 5.5 at 4 °C.

**Crystallization of the PGZL1 Fab variants**. All crystal trials were performed with our Scripps/IAVI/JCSG high-throughput CrystalMation robot (Rigaku) using protein sample (unbound or peptide-bound Fab) at ~ 7 mg/ml. Crystals were obtained using sitting drop vapor diffusion by mixing a 1:1 protein:reservoir solution in 200 nl drops. The MPER$_{671-683}$ peptide sequence used for the Fab complexes was N$_{671}$WFDITNWLWYIK$_{683}$-KKK. Fab-MPER$_{671-683}$ complexes were prepared by mixing Fab with peptide in a 1:5 protein:peptide molar ratio. PGZL1-MPER$_{671-683}$ and H4K3 were co-crystallized with 06:0 PA by mixing highly concentrated protein sample with 06:0 PA (stock solution of 15 mM in 20 mM sodium acetate, pH 5.5) such that the final concentrations of the protein and lipid in the mixture were ~ 7 mg/ml and ~ 8 mM, respectively. The crystallization conditions and cryo-protectant are reported in Supplementary Table 7.

**Data collection, structure determination, and refinement**. X-ray diffraction data sets were collected at SSRL on the 9-2 or 12-2 beamlines (Supplementary Table 7). The data sets were processed using HKL2000[51] or XDS[52]. Phaser[53] was used to find molecular replacement solutions employing the 4E10 Fab variable and constant domains (PDB 2FX7 [https://www.rcsb.org/structure/2fx7]) as the search model. After an initial round of rigid body refinement, model rebuilding was carried out with Coot[54] and refinement with Phenix[55] using different refinement strategies as appropriate for the resolution of each structure. Final statistics are summarized in Supplementary Table 7. Structural images were generated using PyMOL (The PyMOL molecular graphics system).

**MD model of H4K3 interaction with MPER epitope on membrane**. A trimeric model of MPER epitope-gp41 TM region was constructed as previously described[6] using PDB 2MOM [https://www.rcsb.org/structure/2mom] as a template. Briefly, residues 379–403 of LAMP-2A TM region were replaced with the gp41 TM region [residues 686–710; (UNIPROT ID: Q70626, HIV-1-LW123 numbering)], such that Arg796 on gp41 is located in the TM inter-helical interface. The orientation of the MPER in this model is based on the MPER$_{671-683}$ and lipid orientation observed in our crystal structures. The H4K3 Fabs and the 06:0 PA lipids fragments were added to the model by superposing the MPER$_{671-683}$ to the same region of the model. The acyl tails of the crystallographic lipids were extended to the size of a 1,2-dipal-mitoyl-sn-glycero-3-phosphate (DPPA) molecule and the $SO_4$ and $PO_4$ ions bound to CDRH3 and FRL3 regions were replaced with DPPAs with the lipid tails pointing in the same direction as those of the lipids observed in the structures. Thus, the lipid tails are oriented approximately perpendicular to the plane that roughly includes the observed lipid head groups, two anions and Lys683 of the MPER. The lipid bilayer was placed with CHARMM[34] on the putative TM region built to anchor in the membrane. The replacement method was used to place 424 and 455 lipids in the upper and lower leaflet, respectively, of a heterogenous lipid bilayer in a rectangular box of $x = y = 161.2$ Å. The membrane composition of the heterogenous bilayer was chosen based on the HIV-1 membrane lipid composition[56]. Potassium counter ions were placed with the Monte-Carlo method. In the final model, the head group of the lipids corresponding to the sites observed in the crystal structure and the Lys$_{683}$ of MPER are located within the head group region of the membrane outer leaflet.

**Full-length Env sample preparation for cryo-EM**. Recombinant full-length AMC011 Env was expressed in HEK293F cells and purified as described previously[45,57]. Briefly, HEK293F cells (density of $1.6 \times 10^6$ cells/ml; Thermo Fisher Cat#R79007; RRID: CVCL_D603) were transfected with furin and Env-encoding plasmids at 1:3 furin:Env ratio using PEImax. Cells were collected ~ 3 days post transfection, washed once with 400 ml/l cold PBS, and incubated with PGT151 TEV IgG (containing a TEV protease cleavage site between the Fc and the Fab regions) in lysis buffer containing 0.5% v/v Triton X-100, 50 mM Tris pH 7.4, 150 mM NaCl. After removal of cell debris by centrifugation, the supernatant was incubated with Protein A resin (GBiosciences) overnight. Resin was then transferred to gravity flow column, washed, and exchanged to buffer containing 50 mM Tris-HCl pH 7.4, 150 mM NaCl, 0.1% (w/v) DDM (N-Dodecyl-Beta-D-Maltoside), 0.03 mg/ml deoxycholate, and 2 mM EDTA prior to elution by addition of 0.25 mg of TEV protease per liter of initial HEK293F. The complex was further purified using a Superose 6 column (GE Healthcare) using the same buffer excluding EDTA. Purified protein was concentrated to 5.6 mg/ml prior to cryo-EM grid preparation. Three microliters of PGZL1 Fab at 4.5 mg/ml and 1 μl of 1 mM lipid mix (DOPC:DOPS:CHS:PIP2 at 40:40:16:4 molar ratio, Avanti Polar Lipids) was added to 10 μl of purified and concentrated AMC011 Env. Detergent removal was initiated by three additions of ~ 3–5 SM-2 bio beads (Bio-Rad) with 1 h incubation between each addition. After the last incubation, 3 μl of sample was applied to either plasma cleaned 1.2/1.3 C-Flat Holey Carbon grid (Protochips) with 0.5 μl of 0.01% amphiphol A8-35 or to 2/2 Quantifoil Holey Carbon Grid with 0.5 μl of 35 μM LMNG. Amphiphol or LMNG was added directly on grid to improve orientation distribution of particles. Grids were plunge-frozen in liquid ethane using Vitrobot mark IV (Thermo Fisher Scientific) without wait time, blot force of 0 and 7 s blot time.

**Cryo-EM data collection and processing**. Micrographs (5775) were collected using our Titan Krios (Thermo Fisher Scientific) operating at 300 keV and K2 Summit direct electron detector (Gatan). Data were collected with Leginon automated image acquisition software[58] at ×29,000 magnification resulting in pixel size of 1.03 Å in the specimen plane. Forty-six frames were collected for each micrograph with 250 ms exposure time per frame at dose rate of $4.7 \ e^-$/pix/s and with defocus values ranging from −1.0 to −2.5. Frames were aligned and dose weighted with MotionCor2[59] and CTF models for each micrograph were calculated using GCTF[60]. Workflow for subsequent data processing is presented in Supplementary Fig. 8. Particles were picked using DoGPicker[61] and, after initial round of 2D classification in CryoSPARC[62], 356,587 Env particles were moved to Relion[63] for 3D processing. After two rounds of 3D classification, a class of 15,214 Env particles with one PGZL1 Fab bound and two PGT151 Fabs bound showed the best Fab definition and was refined to 8.9 Å resolution according to the FSC 0.143 gold-standard criterion.

**HIV-1 neutralization assays**. Neutralization activity of PG13 donor plasma and monoclonal antibodies was assessed using pseudovirus and a single round of replication in TZM-bl target cells (NIH AIDS Reagent Program; Cat#8129-442, RRID: CVCL_B478). Pseudoviruses were generated by co-transfection of HEK 293T cells (ATCC, Cat#CRL-3216, RRID: CVCL_0063) with an Env-expressing plasmid and an Env-deficient genomic backbone plasmid (pSG3ΔEnv). Pseudo-viruses were collected 48–72 h post transfection and stored at −80 °C. Serial dilutions of plasma/antibody were incubated with virus in presence of DEAE-dextran and the neutralizing activity was assessed by measuring luciferase activity after 48–72 h. Dose–response curves were fitted using nonlinear regression to

determine IC$_{50}$ values. For competition assays, plasma/antibody dilutions were pre-incubated 30 min at RT in the presence or absence of 10 μg/ml of MPER peptide.

**Enzyme-linked immunosorbent assay**. Half-area, 96-well ELISA plates were coated overnight at 4 °C with 50 μL PBS containing 250 ng of antigens per well. The wells were washed four times with PBS containing 0.05% Tween 20 and blocked with 4% non-fat milk (NFM) for 1 h at 37 °C. Serial dilutions of sera/antibodies were then added to the wells and the plates were incubated for 1 h at 37 °C. After washing four times, the wells were treated with goat anti-human IgG Fc conjugated to horseradish peroxidase (HRP) (Jackson ImmunoResearch, Cat#109-035-098), diluted 1:1000 in PBS containing 0.4% NFM and 0.05% Tween 20. The plates were incubated for 1 h at 37 °C, washed four times, and developed by adding HRP substrate diluted in alkaline phosphatase staining buffer (pH 9.8), according to the manufacturer's instructions. The optical density at 405 nm was read on a microplate reader (Biotek Synergy). EC$_{50}$ values were calculated using Prism6 (GraphPad).

**HEp-2 assay**. Antibodies were assayed for autoreactivity using a HEp-2 indirect immunofluorescence kit (Bio- Rad) according to the manufacturer's instructions.

**Antibody binding by biolayer interferometry**. An Octet RED96 system (FortéBio) with BLI was used to assess the binding of PGZL1 and its variants Fab to MPER peptide PDT-081. Biotinylated peptide at 7.5 μg/ml in PBS/0.002% Tween 20/0.01% bovine serum albumin was captured on the surface of Streptavidin biosensors (FortéBio) for 8 min. The biosensor was exposed to a serial dilution of Fabs for 5 min and then to buffer for 5 min, to acquire association and dissociation sensograms, respectively. $K_D$ values were calculated as $k_{off}/k_{on}$ based on five sensograms from the dilution series with a minimum $R^2$ value of 0.99. The sensograms were corrected using the blank reference and fitting was accomplished using the FortéBio Data Analysis 7 software package.

**Flow cytometry**. Comb-mut V4 cell line was generated and characterized as described[25]. Following a similar protocol, the MPER-TM$_{654-709}$ cell line was developed. Briefly, a PCR amplicon encoding a TPA leader sequence, codon-optimized MPER-TM (654-709) of BG505, followed by a stop codon was cloned into NotI and XhoI sites of pLenti-III-HA (Applied Biological Materials). The resulting MPER-TM lentiviral vector was used to generate lentiviral particles, which were then used to transduce 293T cells following the manufacturer's instructions. The transduced cells were cultured in medium containing 10 μg/ml puromycin to select for stable integrants and were sorted on a BD Aria flow cytometer to select the 10E8 (high) stable cell line MPER-TM$_{654-709}$.

For flow cytometry experiments, a total of $10^7$ cells of stable cell lines HIV-1 MPER-TM$_{654-709}$ or Comb-mut Env (V4) were washed in PBS and labeled with Fixable Aqua Dead Cell Stain (Life Technologies). Cells were washed in FACS buffer (PBS supplemented with 2% heat-inactivated fetal bovinse serum) and were stained with monoclonal antibody. After another wash, cells were stained using APC-conjugated mouse anti-human Fc (BioLegend HP6017). Soluble CD4 was incubated with cells for 30 min prior to staining cells with antibody. Cells were acquired and analyzed by using NovoCyte (ACEA Biosciences). Data were analyzed using FlowJo software (Tree Star).

**BN-PAGE mobility shift assay**. Virus samples were pre-incubated with Fab fragments of antibodies for 30 min at RT. Samples were then solubilized with 1% DDM for 20 min on ice. Env was separated using BN-PAGE and was detected by western blotting using a cocktail of gp120 and gp41 antibody probes. Western blottings were imaged using a Chemidoc XRS (Bio-Rad) and analyzed using Image Lab software (Bio-Rad). The center of intensity for each Env trimer band and the distance that it had migrated along the gel was calculated. The relative shift of each band was calculated by setting the maximum shifted band in each experiment to 1, which corresponds to threefold occupancy of Fab per trimer and the antibody-free control to 0. The trimer occupancy at each Fab concentration was determined by calculating the relative shift for control Fabs PGT126 (3 Fabs/trimer), PGT151 (2 Fabs/trimer), and PG9 (1 Fab/trimer).

**Site-directed mutagenesis**. Mutagenesis was performed using a Quikchange site-directed mutagenesis kit (Agilent Technologies).

**Lipid insertion propensity**. Lipid insertion propensity scores were calculated using the MPEx (Membrane Protein Explorer) software as the sum of ΔGwif, the free energy of transfer of an amino acid from water to POPC (1-palmitoyl-2-oleoyl-glycero-3-phosphocholine) interface, over all amino acids of the antibody variable heavy and light domains.

**Statistical analysis**. For all mAb/serum pseudovirus neutralization and ELISA assays, the IC$_{50}$ or concentration of mAb/dilution of serum needed to obtain 50% neutralization against a given pseudovirus was calculated from the linear regression of the linear part of the neutralization curve. For neutralization assays in which a

fold change in IC$_{50}$ imparted by a virus mutant or virus treatment was reported, the IC$_{50}$ obtained for one virus/assay condition was divided by the IC$_{50}$ obtained for the other virus/assay condition, as indicated in the figure legends. Two-way analysis of variance was used for multiple group comparison.

**Full-length Env amplification sequencing and analysis.** HIV-1 Env was sequenced as described[49]. Briefly, virions were purified from plasma through a sucrose cushion and ultracentrifugation. RNA was extracted (Viral RNA Mini Kit, Qiagen) and reverse transcribed (SuperScript III, Thermo Fisher). PCR was performed with subtype B primers using 45 PCR cycles. Four replicate PCR reactions were pooled, purified (QIAquick, Qiagen), visualized, and quantified (2100 Bioanalyzer System, Agilent Biosciences). Preparation and sequencing of SMRTbell template libraries of ~ 2.6-kb insert size were performed according to the manufacturer's instructions (Pacific Biosciences) using P6/C4 chemistry on the RS-II.

CCS sequences were constructed using the PacBio SMRTportal software (version 2.3). The Robust Amplicon Denoising algorithm[64] was used for error correction and MAFFT[65], with manual curation, was used to construct a multiple sequence alignment. Phylogenies were reconstructed using FastTree v2.1 and were visualized with FigTree. Geno2Pheno 2.5[66] was used to predict co-receptor tropism.

**Reporting summary.** Further information on research design is available in the Nature Research Reporting Summary linked to this article.

## Data availability

The PGZL1 HC and LC variable region sequences have been deposited into Genbank, accession MK497833–MK497838. The atomic coordinates and structure factors of PGZL1 variants have been deposited in the Protein Data Bank, with accession codes: 6O3D (PGZL1); 6O3G (PGZL1-MPER$_{671-683}$); 6O3J (PGZL1-MPER$_{671-683}$-06:0 PA); 6O3K (H4K3); 6O3L (H4K3-MPER$_{671-683}$); 6O3U (H4K3-06:0 PA); 6O41 (PGZL1 gVmDmJ-Protein G); and 6O42 (PGZL1 gVmDmJ-MPER$_{671-683}$-06:0 PA). The cryo-EM reconstruction of full-length AMC011-PGT151-PGZL1 complex has been deposited in the Electron Microscopy Data Bank with accession code EMD-0620. Env sequences and browser-based visualizations are available at https://flea.ki.murrell.group/view/PG13/sequences/. Datasets generated during and/or analyzed during the current study are included in the supplementary Source Data file.

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

## Acknowledgements

We thank Katie Bauer, Harry Quendler, Alex Ramirez, Yuanzi Hua, and Henri Tien for technical help. We thank Dr Netanel Tzarum for advice on BLI experiments. This work was supported by NIH R01 AI114401 and AI143563 to M.B.Z.; by Scripps CHAVI-ID (UM1 AI100663), Scripps CHAVD (UM1 AI144462), and HIV Vaccine Research and Design (HIVRAD) program (P01 AI124337) to J.Z.; by NIH R01 Grants AI129698 and AI140844 to J.Z.; by James B. Pendleton Charitable Trust; and by IAVI with support of (United States Agency for International Development) USAID, Ministry of Foreign Affairs of the Netherlands, and the Bill & Melinda Gates Foundation (CAVD OPP1084519). For a list of IAVI donors, see www.iavi.org. Use of the Stanford Synchrotron Radiation Lightsource, SLAC National Accelerator Laboratory, is supported by the U.S. DOE, Office of Science, Office of Basic Energy Sciences (DE-AC02-76SF00515). The SSRL Structural Molecular Biology Program is supported by DOE Office of Biological and Environmental Research, and by NIH, NIGMS (P41GM103393). We thank the donor who enabled this study. Contents of this study are solely the author's responsibility and may not reflect views of the NIH, USAID, or the US Government.

## Author contributions

Conceptualization: L.Z., A.I., J.Z., I.A.W., and M.B.Z. Methodology: L.Z., A.I., L.H., and K.R. Investigations: L.Z., A.I., L.H., K.R., D.P.L., E.L., D.S., A.S., T.V., and A.S.K. Resources: M.B.Z., I.A.W., J.Z., A.B.W., D.R.B., P.P., and P.G.I. Writing: L.Z., A.I., J.Z., I.A.W., and M.B.Z. Supervision: M.B.Z., I.A.W., J.Z., A.B.W., and B.M. Funding: M.B.Z., I.A.W., D.R.B., A.B.W., and J.Z.

## Competing interests

The authors declare the following competing interests: A U.S. provisional patent application entitled "Human broadly neutralizing antibodies against the membrane-proximal external region of HIV gp41" is to be filed covering the manuscript in its entirety, with the currently named inventors L.Z., A.I., J.Z., I.A.W., and M.B.Z., all of whom are currently employed by or affiliated with The Scripps Research Institute. All other authors declare no competing interests.

## Additional information

## IAVI Protocol G Investigators

George Miiro[10], Jennifer Serwanga[10], Anton Pozniak[11], Dale McPhee[12], Oliver Manigart[13], Lawrence Mwananyanda[13], Etienne Karita[14], André Inwoley[15], Walter Jaoko[16], Jack DeHovitz[17], Linda-Gail Bekker[18], Punnee Pitisuttithum[19], Robert Paris[20] & Susan Allen[21]

[10]MRC/UVRI Uganda Research Unit on AIDS, Uganda Virus Research Institute, Entebbe, Uganda. [11]St Stephens AIDS Trust, Chelsea and Westminster NHS Foundation Trust, London, UK. [12]NRL, St Vincent's Institute, Melbourne, Victoria, Australia. [13]Zambia Emory HIV Research Project, Lusaka, Zambia and Rwanda-Zambia HIV Research Group, Emory University, Atlanta, GA, USA. [14]Projet San Francisco, Kigali, Rwanda and the Rwanda-Zambia HIV Research Group, Emory University, Atlanta, GA, USA. [15]CeDReS/CHU Treichville, Abidjan, Côte d'Ivoire. [16]Kenya AIDS Vaccine Initiative, College of Health Sciences, University of Nairobi, Nairobi, Kenya. [17]SUNY Downstate Medical Center, Brooklyn, New York, USA. [18]Desmond Tutu HIV Centre, University of Cape Town, Cape Town, South Africa. [19]Faculty of Tropical Medicine, Mahidol University, Bangkok, Thailand. [20]Department of Retrovirology, Armed Forces Research Institute of Medical Sciences, Bangkok, Thailand. [21]Rwanda-Zambia HIV Research Group, Emory University, Atlanta, Georgia, USA

