## [Peer Review File · Nature Communications]

Reviewers' Comments:

Reviewer #1:

Remarks to the Author:

The manuscript by Zhang et al. entitled "MPER Antibody Neutralizes HIV Using Germline Features Shared among Donors" describes a new MPER-directed 4E10-like neutralizing antibody, PGZL1, from an HIV-infected individual. PGZL1 shares the germline V/D-region genes with 4E10, however with IgG1 subclass and a shorter CDRH3, and lacks polyreactivity. The neutralization breadth of PGZL1 is 84%, with potency lower than 4E10, assessed with a 130-virus panel. Using deep sequencing, the authors were able to identify a PGZL1 sublineage, with variant possessing 100% virus coverage and potency similar to 4E10. A germline version of PGZL1 neutralizes 12% of viruses, retaining MPER affinity, in contrast to germline versions of other MPER bnAbs identified previously, which typically have no detectable binding to MPER. The discovery of PGZL1 lineage suggests that it is possible to elicit such B cell lineages with well-designed immunogens. Furthermore, the authors obtained a panel of structures of PGZL1-MPER-lipid complex, which provides details of how PGZL1 interact with the lipid membrane and the MPER. Finally, a comprehensive model of PGZL1 interaction with Env including MPER in the context of viral membrane was generated using the crystal structural information and the cryo-EM 3D reconstruction of Env of isolate AMC011 in complex with PGZL1 and PGT151. The detailed genetic and structural analysis of this newly identified MPER-directed bnAb lineage performed in this manuscript will be quite informative for future immunogen design to re-elicite cognate bnAb responses. Together with the recent related publication by Krebs et al., 2019, *Immunity* 50, 677–691, this manuscript highlights the rationale of 4E10 lineage-based vaccine design, a very strong message for the field. This manuscript is written and thought out very well. Some suggestions are listed as below for strengthening purpose.

Major point

The statement of "PGZL1 gVmDmJ binds directly to native Env and neutralizes some primary isolates" needs to be clarified. The virus washout experiment (Table S6) did likely suggest PGZL1 gVmDmJ binding with the virion prior to the contact with target cell (receptor CD4 etc.). However, it is not clear whether the binding is definitely with the native Env or just the viral membrane, or both. In the Env cell surface staining experiment (Fig. 2D), there is virtually no detectable difference between PGZL1 gVmDmJ binding to the Env expressed on the cell surface and the 293 cell surface alone, which does not support the notion that PGZL1 gVmDmJ binds the native Env directly. Therefore, such statement could be revised to be consistent with the experiment data, such as "PGZL1 gVmDmJ is able to bind the virions of some primary isolates weakly prior to the receptor engagement..."

Minor points

1. Table S1, the competitor peptide concentration is listed as 10 mg/ml, which is high and maybe toxic to the cells. Would 10 ug/ml be the working concentration?
2. Page 6, Figure S3C was cited in the first paragraph on correlation between PGZL1 and H4K3 IC50s. Should Figure S3D be cited instead? The same citation in the 3rd paragraph on page 7.
3. Page 8, in the first line "BJOX025, HxB2 and 92TH021" were mentioned in Table S6. But "BJOX025" was not listed in Table S6. It should be added to the table or omitted from the text.
4. Methods part, please indicate if the MPER peptide PDT-081 was biotin-labeled at the N-terminus? It seems to be biotin-labeled to couple with Sa-PE or-APC.
5. Figure 2C, the Y-axis label is "nM", which can be changed to "BLI Response (nM)" to be clear to the readers.
6. Figure S5A, please specify what is the "X" in the 4th sequence entry. It would help to show the MPER sequence of COT6 in this panel as well.

Reviewer #2:

Remarks to the Author:

Zhang et al. have analyzed an MPER-directed bnAb, called PGZL1, and its variants from an HIV-1 infected patient. These antibodies share the same V/D-region germline genes as 4E10 but are of the more major IgG1 class, have shorter CDRH3 loops and are less polyreactive than 4E10. The authors performed a variety of experiments (BLI, neutralization, etc.) to show how effective PGZL1 and its variants are in binding (low nanomolar range) and neutralizing viruses and their Envs. The authors also report a series of structures of PGZL1 and several variants, either alone or in complex with other Env constructs or lipids to give insight into Env recognition and viral resistance.

The crystal structures of the inferred germline with and without the MPER peptide show how the interaction could change during virus evolution. The crystal structures with PGZL1 also revealed that PGZL1 is very similar structurally to 4E10. By analysis of mutants, the authors show that the paratope and MPER binding site for both classes of antibodies are similar. This suggests that antibodies with similar mechanisms (yet of different class) could be elicited by proper immunogens and the structural information with germline variants would be helpful for this as well. Of particular note, the authors did show that the germline revertants can bind and neutralize HIV Env, which is an important property when considering vaccine design opportunities and would therefore be of interest to the HIV vaccine design community.

There are still some questions that arise based on the work described and some of the experiments performed. These are outlined below along with some suggestions, which should be addressed to strengthen the conclusions being made.

Major comments:

Krebs et al. published a paper in *Immunity* in 2019 on three MPER-directed antibody lineages, some of which are of the IgG1 class (and are less autoreactive than 4E10). Comparisons should be done between PGZL1 and the variants and these other MPER antibodies, in sequence, gene usage, structure, etc.

How long did it take for the donor to produce PGZL1? It was mentioned in the text that three blood draws were done within a 9-month period, but it was unclear as to what the relationship is of that 9-month period to the time of infection. Moreover, PGZL1 and the variants seem to have high levels of SHM – is it worth going after such bnAbs if there are others that have lower levels of SHM (e.g. the ones described by Krebs et. al)?

Has the serum from 5 months prior to the retrieval of PGZL1 been tested against the same panel of viruses as H4K3? If not, it should be and compared to the neutralization properties of H4K3 to show that such a chimeric antibody is biologically relevant.

While crystal structures were obtained of PGZL1 and H4K3 with lipids, how can one be sure that the lipids in the crystal structure are biologically relevant? Perhaps some binding experiments could be done with mutants and the lipids to ensure that the sites of interaction indicated by the crystal structure are valid (and specific for those sites).

Other comments:

Figures and Figure Legends:

Figure 1: panels A and E – The units of the numbers in the table should be indicated

Figure 2: k_{on} and k_{off} should have lowercase k 's in the figure legend. It would be helpful to have the data for PGZL1 in panel F for comparison.

Figure 4: panel B – is the structure of 4E10 that is superposed of an unliganded Fab?

Figure 5. panel G – Is the arrow supposed to be there? If so, what is it pointing to? As its drawn it appears to be pointing to the phospholipid model.

Supplementary Fig. 4. It would be helpful to have the numbers listed for the portion of the antibody sequences that are shown so that it is easy to refer to specific amino acids as they are discussed in the text.

Supplementary Fig. 5. A scale bar is missing from the viral lineage tree (unless it's not drawn to scale, in which case that should be indicated).

Minor points and questions:

Page 6 – The sentence "Overall, H4K3 is exceptionally broad and equipotent with 4E10 that is protective against HIV-1 in animal models" is confusing and I am not sure what the reference to animal models is referring to.

Page 7 – ADA-CM needs to be defined.

Were other positions mutated to Ala other than the ones mentioned on page 10? If so, it should be indicated (even if those mutations had no effect).

On page 12, before the section on the cryo-EM structure, it should be indicated that the model was obtained using MD simulations. It wasn't clear until the following results section. Also, where is K683 in the corresponding figure (panel 5G)?

Page 13 – the sentence should read "Through crystallography and cryo-EM, we have determined the orientation, angle of approach ..."

Reviewer #3:

Remarks to the Author:

Zhang and colleagues report on a novel, broadly neutralising MPER antibody (PGZL1) that shares many features with mAb 4E10. Importantly, PGZL1 shares germline V/D-region genes with 4E10 (V-region genes VH1-69 and VK3-20, as well as D-region D3-10), but has a shorter CDRH3 and lower poly-reactivity. To provide insights into how PGZL1 variants interact with MPER on the viral membrane, the authors conducted a detailed structural analysis of lipid-bound PGZL1 variants and performed cryo-EM reconstruction of Env AMC011 in complex with PGZL1.

The authors highlight several notable features of PGZL1: (i) PGLZ1 exhibits high breadth (84% neutralisation against a panel of 130 different viruses; 100% for variant H4K3); (ii) PGZL1 is intriguingly an IgG1, in contrast to 4E10 and many other MPER neutralising antibodies (which were isolated as IgG3); and (iii) the PGZL1 CDRH3 is three residues shorter, and uses a different J gene, compared to 4E10. The authors also emphasise that a PGZL1 germline revertant (that contains the mature PGZL1 CDR3s) still retains considerable neutralising activity.

Major points of criticism:

The study is mostly very thorough and elegantly done. It would, however, be a beautiful paper if it were not for a severe omission. The authors present PGZL1 as if it would be the very first 4E10-like antibody that has been discovered. In truth, Krebs et al (Immunity 2019; DOI:10.1016/j.immuni.2019.02.008) recently reported on three lineages of MPER antibodies in a single donor. The lead antibody discovered by Krebs et al (VRC42) is also an IgG1 MPER-specific Ab that uses VH1-69 and VK3-20. The original VRC 42 clone appears to have a higher breadth than the original PGZL1. Similar to what is seen for PGZL1, Krebs et al. also report that a germline revertant of VRC34 retains the ability to engage the MPER antigen. Thus, all key findings reported here by Zhang et al. with PGZL1 have also been observed for VRC34.

It is not easy to comprehend why, knowing of the competing study (Krebs et al. was published online March 12), the authors did not adapt their manuscript before submission. Given the similarities between PGZL1 and VRC42, analysis of PGZL1 must also include VRC42. This includes all parts of the manuscript, sequence alignments (Fig S1), phylogenetic analyses (Fig 1c), and Table S2 to name but a few. The entire introduction and discussion must also be adapted to highlight where PGZL1 may be superior to VRC42 and/or where the analysis of PGZL1 adds to the understanding of MPER reactive antibodies that goes beyond that shown by Krebs et al. Without these points being addressed, the importance of the findings by Zhang and colleagues cannot be rated.

Other comments:

Patient/sampling information is missing. For how long has the donor been infected? Were the 3 time points used for NGS before or after the date of Ab isolation?

Fig. 2F: Why is breadth at 200ug/ml reported, as opposed to at 50ug/ml in Fig 1?

"The IC50s of PGZL1 and H4K3 are tightly correlated ($r=0.82$, $p<0.0001$) suggesting a similar neutralisation mechanism (Figure S3C)." This refers to fig S3D not S3C.

"The IC50s of PGZL1 and PGZL1 gVmDmJ correlated modestly ($r=0.52$, $p=0.0013$) and some isolates were hypersensitive to the revertant (Figure S3C)." What do the authors define as hypersensitive? Table S5 does not show any virus with lower IC50 against the revertant than the wt Ab.

Since the authors did not succeed in isolating infection competent Env clones from the PGZL1 donor, the data on PG13 Env and the putative resistance mutation should probably be dropped. This does not add to the paper without functional confirmation. Sticking with the COT6 data is cleaner.

The authors state in the abstract that PGZL1 lacks polyreactivity, but conclude in the context of Figure 3: "Overall, PGZL1 antibodies showed less polyreactivity than 4E10". Less certainly does not mean lacking. This needs to be corrected in the abstract.

More data on the patient plasma would have been nice to see. Does PGZL1 capture the breadth of the plasma?

The data on the germ line revertant are difficult to interpret as not all variants have been tested in all assays. In Fig 2B, PGZL1gVgDgJ and 4E10 gVgDgJ need to be shown. All three variants of 4E10 (wt., 4E10 gVmDMJ and 4E10 gVgDgJ) need to be shown in Fig 2C. 4E10 gVmDmJ_L100cF , as it has improved neutralisation capacity according to Figure S4C, should also be included in these binding experiments.

We would like to thank the reviewers for helpful comments and excellent suggestions that have allowed us to modify and greatly strengthen our manuscript. Please find below a point-by-point response to the reviewer comments.

Reviewer#1

The manuscript by Zhang et al. entitled “An MPER Antibody Neutralizes HIV-1 Using Germline Features Shared among Donors” describes a new MPER-directed 4E10-like neutralizing antibody, PGZL1, from an HIV-infected individual. PGZL1 shares the germline V/D-region genes with 4E10, however with IgG1 subclass and a shorter CDRH3, and lacks polyreactivity. The neutralization breadth of PGZL1 is 84%, with potency lower than 4E10, assessed with a 130-virus panel. Using deep sequencing, the authors were able to identify a PGZL1 sublineage, with variant possessing 100% virus coverage and potency similar to 4E10. A germline version of PGZL1 neutralizes 12% of viruses, retaining MPER affinity, in contrast to germline versions of other MPER bnAbs identified previously, which typically have no detectable binding to MPER. The discovery of PGZL1 lineage suggests that it is possible to elicit such B cell lineages with well-designed immunogens. Furthermore, the authors obtained a panel of structures of PGZL1-MPER-lipid complex, which provides details of how PGZL1 interact with the lipid membrane and the MPER. Finally, a comprehensive model of PGZL1 interaction with Env including MPER in the context of viral membrane was generated using the crystal structural information and the cryo-EM 3D reconstruction of Env of isolate AMC011 in complex with PGZL1 and PGT151. The detailed genetic and structural analysis of this newly identified MPER-directed bnAb lineage performed in this manuscript will be quite informative for future immunogen design to re-elicite cognate bnAb responses. Together with the recent related publication by Krebs et al., 2019, Immunity 50, 677–691, this manuscript highlights the rationale of 4E10 lineage-based vaccine design, a very strong message for the field. This manuscript is written and thought out very well. Some suggestions are listed as bellow for strengthening purpose.

We thank Reviewer #1 for the encouraging remarks and suggestions including:

“The detailed genetic and structural analysis of this newly identified MPER-directed bnAb lineage performed in this manuscript will be quite informative for future immunogen design... ..this manuscript highlights the rationale of 4E10 lineage-based vaccine design, a very strong message for the field. This manuscript is written and thought out very well.”

We address below the reviewer’s suggestions and comments:

1. *The statement of “PGZL1 gVmDmJ binds directly to native Env and neutralizes some primary isolates” needs to be clarified. The virus washout experiment (Table S6) did likely suggest PGZL1 gVmDmJ binding with the virion prior to the contact with target cell (receptor CD4 etc.). However, it is not clear whether the binding is definitely with the native Env or just the viral membrane, or both. In the Env cell surface staining experiment (Fig. 2D), there is virtually no detectable difference between PGZL1 gVmDmJ binding to the Env expressed on the cell surface and the 293 cell surface alone, which does not support the notion that PGZL1 gVmDmJ binds the native Env directly. Therefore, such statement could be revised to be consistent with the experiment data, such as “PGZL1 gVmDmJ is able to bind the virions of some primary isolates weakly prior to the receptor engagement...”*

We acknowledge the point made here and have revised the text (Page 8), which currently states “Thus, PGZL1 gVmDmJ binds weakly to and neutralizes some primary isolates prior to receptor engagement.” We note that, in addition to the virus antibody washout data, we also presented data showing that PGZL1 gVmDmJ strongly and specifically stains cells displaying MPER-TM (Fig. 2D, left), cells bearing Env treated with soluble CD4, as well as data showing that PGZL1 gVmDmJ can shift the mobility of the detergent-extracted native trimer in BN-PAGE showing that PGZL1 gVmDmJ can bind the MPER displayed on the cell surface, cell surface Env after solubilized CD4 addition and detergent extracted Env. These data, and the fact that upon washout some but not all viral isolates were (weakly) neutralized by PGZL1 gVmDmJ, suggests that PGZL1 gVmDmJ can bind weakly to the native Env trimer of the former and not the latter viral isolates. Note that because the native Env is in the membrane, we cannot formally distinguish in this case between native trimer in the presence and absence of membrane since the presence of membrane is included in the definition of native Env on virions.

2. *Table S1, the competitor peptide concentration is listed as 10 mg/ml, which is high and maybe toxic to the cells. Would 10 ug/ml be the working concentration?*

We thank the reviewer for pointing out the typo. The concentration has been corrected to 10 µg/ml.

3. *Page 6, Figure S3C was cited in the first paragraph on correlation between PGZL1 and H4K3 IC50s. Should Figure S3D be cited instead? The same citation in the 3rd paragraph on page 7.*

The text in question has been corrected (Page 6 and 7), and we thank the reviewer for pointing this out.

4. *Page 8, in the first line “BJOX025, HxB2 and 92TH021” were mentioned in Table S6. But “BJOX025” was not listed in Table S6. It should be added to the table or omitted from the text.*

BJOX025 was incorrectly written and should have been BJOX025000, which is listed in Table S6. We corrected the text (Page 7 and Table S6) to correct this typo.

5. *Methods part, please indicate if the MPER peptide PDT-081 was biotin-labeled at the N-terminus? It seems to be biotin-labeled to couple with Sa-PE or-APC.*

The text has been updated (Page 15). The MPER peptide PDT-081 was biotin-labeled; however, we note that the biotin was attached to the C-terminus, not the N-terminus.

6. *Figure 2C, the Y-axis label is “nM”, which can be changed to “BLI Response (nM)” to be clear to the readers.*

The Y-axis in Figure 2C has been re-labeled as the reviewer suggests.

7. *Figure S5A, please specify what is the “X” in the 4th sequence entry. It would help to show the MPER sequence of COT6 in this panel as well.*

The X in the MPER sequence is a stop codon. The Env is thus non-functional, but we disfavor discarding such sequences, as they can provide evolutionary clues. Thus, we have also noted that 'X' is a stop codon in the figure caption (current Figure S6A).

We thank the reviewer for the helpful comments that have led us to improve the manuscript.

Reviewer #2

Zhang et al. have analyzed an MPER-directed bnAb, called PGZL1, and its variants from an HIV-1 infected patient. These antibodies share the same V/D-region germline genes as 4E10 but are of the more major IgG1 class, have shorter CDRH3 loops and are less polyreactive than 4E10. The authors performed a variety of experiments (BLI, neutralization, etc.) to show how effective PGZL1 and its variants are in binding (low nanomolar range) and neutralizing viruses and their Envs. The authors also report a series of structures of PGZL1 and several variants, either alone or in complex with other Env constructs or lipids to give insight into Env recognition and viral resistance.

The crystal structures of the inferred germline with and without the MPER peptide show how the interaction could change during virus evolution. The crystal structures with PGZL1 also revealed that PGZL1 is very similar structurally to 4E10. By analysis of mutants, the authors show that the paratope and MPER binding site for both classes of antibodies are similar. This suggests that antibodies with similar mechanisms (yet of different class) could be elicited by proper immunogens and the structural information with germline variants would be helpful for this as well. Of particular note, the authors did show that the germline revertants can bind and neutralize HIV Env, which is an important property when considering vaccine design opportunities and would therefore be of interest to the HIV vaccine design community.

There are still some questions that arise based on the work described and some of the experiments performed. These are outlined below along with some suggestions, which should be addressed to strengthen the conclusions being made.

We thank the reviewer for his positive comments on the manuscript.

Please find below the responses to Reviewer #2's comments.

- 1. Krebs et al. published a paper in Immunity in 2019 on three MPER-directed antibody lineages, some of which are of the IgG1 class (and are less autoreactive than 4E10). Comparisons should be done between PGZL1 and the variants and these other MPER antibodies, in sequence, gene usage, structure, etc.*

We now include a comparison between PGZL1 antibodies and the relevant 4E10-like VRC42 MPER antibody lineage that was described recently by Krebs et al. Indeed, there are important similarities between these 4E10-like antibodies, *i.e.*, PGZL1, VRC42 and 4E10, which is notable as they come from three unrelated donors from South Africa, Thailand, and Europe, who were infected with a clade B, clade AE and a clade B virus, respectively. We are very enthusiastic about these similarities as they show that 4E10-like antibodies are not so rare, and suggest they may be elicitable by vaccination. Comparisons of VRC42 and PGZL1 antibodies have been now added including phylogenetic analysis (Figure 1C), sequence alignment (Figure S1A), hydrophobicity plots (Figure S1B), gene usage (Table S2) and structure (Figure S5). These comparisons strengthen the paper by showing: (i) how these

4E10-like antibodies evolved from similar V_H , V_K and D_H germline genes and three different J-region germline genes into antibodies with differences in V_H/V_K SHMs that can perform a very similar function; (ii) 4E10-like antibodies contain hydrophobic residues in similar positions in their CDRH3 with (iii) some similar residues in N2 addition regions in CDRH3; (iv) structural comparison reveals slight differences in the binding of the N-term of the MPER epitope (residues 671-675) perhaps due to different residues from the different J gene usage (F100g in PGZL1, H4K3 and 4E10 and M100g in VRC42.01).

We thank the reviewer for prompting us to include comparisons with the 4E10-like antibody from Krebs et al., which has improved the quality and the relevance of our manuscript to vaccine design.

- 2. How long did it take for the donor to produce PGZL1? It was mentioned in the text that three blood draws were done within a 9-month period, but it was unclear as to what the relationship is of that 9-month period to the time of infection. Moreover, PGZL1 and the variants seem to have high levels of SHM – is it worth going after such bnAbs if there are others that have lower levels of SHM (e.g. the ones described by Krebs et al.)?*

Time-of-infection data are unfortunately not available with the PGZL1 donor as only a few precious samples were collected. (Protocol G was a cross-sectional, not a longitudinal study to screen for broad neutralizing activity in donor plasma). We state in the manuscript that the PGZL1 antibody was obtained by single B cell cloning from Visit #5, which occurred 9 months after the patient's first visit; however, by that time, the patient was already infected for an unknown period of time, and samples from that visit or earlier time points are not available. Indeed, it is possible that the donor was infected for a long time, given that PGZL1 has a high level of SHM (20.9% in V_H), which is typical for bnAbs identified during chronic HIV infection. The recombinant variant H4K3 came from a sample five months (Visit #2) prior to PGZL1 and has less SHM (17.6%).

Is it worth going after more heavily mutated antibodies? BnAbs with low SHM are preferable for lineage-informed vaccine design, which is why we generated the PGZL1 gVmDmJ and gVgDgJ revertants that lack SHM in V_H/V_K . Although these revertants are somewhat artificial, they are significant for vaccine design in that these revertants can still bind to the MPER. Furthermore, our analyses using phylogenetics, mutagenesis and structure-function studies of PGZL1, along with a comparison with VRC42 and its UCA, reveal many important facts and features about 4E10-class bnAbs. For example: (i) they can contain diverse SHM, as well as different germline J_H genes, and different CDRH3 lengths, which might make 4E10-like bnAbs easier to elicit; (ii) CDRH3 is crucial for binding to the MPER in the context of membrane lipid. Since we do not know which path might elicit 4E10-like by vaccination, it is critical to understand the germline genetic and structural bases of neutralization by bnAbs from different donors, so that Env immunogens can be developed based on elements shared among different 4E10-like lineages. This will be a focus for follow-up studies.

- 3. Has the serum from 5 months prior to the retrieval of PGZL1 been tested against the same panel of viruses as H4K3? If not, it should be and compared to the neutralization properties of H4K3 to show that such a chimeric antibody is biologically relevant.*

As we stated above, samples were extremely limited from the PGZL1 donor; however, we were able to test the plasma sample from Visit 2 (5 months prior to the retrieval of PGZL1) against the 6-virus panel together with H4K3 and included these results in the manuscript (Figure 1A, and page 4). We do not currently understand why 92TH021 is not neutralized by

the sera from visit #2. However, we point out that representation in plasma is not the only measure of potential biological relevance of H4K3. For example, H4K3 shows: (i) H4K3 is biologically relevant in the sense that its heavy (H4) and light (K3) chains were identified from the donor repertoire, suggesting that they were present during the PGZL1 lineage development. The extraordinary breadth of H4K3 indicates that coordinated H/L co-evolution is also most likely critical to the functional development of a bNAb lineage. (ii) another antibody can achieve similar potency and breadth as 4E10 (thus greater potency and breadth than VRC42.01) with three fewer residues in CDRH3, and with less SHM than PGZL1; (iii) a phylogenetically distant sub-lineage of PGZL1 evolved paratope elements with the capacity for extremely broad HIV-1 neutralization. Moreover, it is currently unknown whether and which 4E10-like antibodies may be elicitable by vaccination, so vaccine design should consider interactions of all 4E10-like antibodies and Env to try to elicit the more potent bnAbs in appropriate animal models. We note that H4K3 could also be an important lead for therapy.

4. *While crystal structures were obtained of PGZL1 and H4K3 with lipids, how can one be sure that the lipids in the crystal structure are biologically relevant? Perhaps some binding experiments could be done with mutants and the lipids to ensure that the sites of interaction indicated by the crystal structure are valid (and specific for those sites).*

To answer the reviewer's comment, we have obtained new data that addresses the biological relevance of the lipid binding sites observed in the PGZL1 variant antibody H4K3, and have included these data in the revised manuscript. We designed three mutants:

- 1) H4K3_SFS28EPE that contain the CDRH1 lipid binding site residues $S_{28}F_{29}S_{30}$ mutated to $E_{28}P_{29}E_{30}$ in the heavy chain and the CDRH3 $R_{100b} - R_{100e}$ site disrupted by G50D mutation in the light chain;
- 2) H4K3_RS73PE that contain the FRH3 lipid binding site residues $D_{72}R_{73}S_{74}$ mutated to $D_{72}P_{73}E_{74}$, in the heavy chain and the CDRH3 $R_{100b} - R_{100e}$ site disrupted by G50D mutation in the light chain;
- 3) H4K3_5M that has all the mutations included in the previous two mutants, namely SFS28EPE/RS73PE/G50D;

The proline mutation was used to impede the interaction between the main-chain nitrogen atoms on H4K3 and the lipid PO_4 moieties, whereas the glutamate mutations were used to introduce charge repulsion with the lipid bilayer.

ELISA and BLI binding experiments showed that the mutants retain similar binding affinity for the MPER peptide in the absence of the viral membrane, suggesting that the mutations do not affect binding to MPER. Neutralization experiments with these mutants, however, show that the potency was diminished by ~7 to 84-fold, suggesting that the interaction with the membrane lipid head groups in these mutants was compromised.

Additionally, we would like to emphasize that the CDRH1 lipid-binding site was also observed in 4E10 (see our previous paper in *Immunity*). The position of the lipid binding sites in H4K3 and PGZL1, and the orientation of these Fabs with regard to the MPER-viral membrane epitope is consistent with our previous results with 4E10 and 10E8 antibodies (see our *Immunity* and *PLoS Pathogens* papers). Moreover, this orientation is also confirmed by the cryo-EM reconstruction of Env-PGT151-PGZL1 complex. Thus, the recurrence of lipid binding sites in H4K3, PGZL1 and 4E10, as well as the agreement between the membrane model based on these lipids and the position of the membrane based on our cryo-EM studies of PGZL1-bound Env, strongly suggests specificity and biological relevance.

5. *Figure 1: panels A and E – The units of the numbers in the table should be indicated*

The units have been included in the tables.

6. *Figure 2: k_{on} and k_{off} should have lowercase k 's in the figure legend. It would be helpful to have the data for PGZL1 in panel F for comparison.*

The figure legend has been amended and the PGZL1 data were included in panel F for comparison.

7. *Figure 4: panel B – is the structure of 4E10 that is superposed of an unliganded Fab?*

The Figure 4, panel B shows superposition of the PGZL1 - MPER₆₇₁₋₆₈₃ complex (green-HC, wheat-LC, red MPER) with 4E10 - MPER₆₇₁₋₆₈₃ (gray). The figure legend was updated to make this clear.

8. *Figure 5. panel G – Is the arrow supposed to be there? If so, what is it pointing to? As its drawn it appears to be pointing to the phospholipid model.*

Yes, the arrow in Fig. 5, panel G was intentional and was added to symbolize the flow of reasoning behind our structural models. Thus, based on the initial experimental information regarding the lipid and anion binding sites on H4K3 (to the left of the arrow), we constructed a model of H4K3 bound to MPER-viral membrane lipid-composite epitope (to the right of the arrow). The arrow means that the model is based on the experimental structural data on lipids and anions bound to the Fab. We modified the Figure legend to clarify this.

9. *Supplementary Fig. 4. It would be helpful to have the numbers listed for the portion of the antibody sequences that are shown so that it is easy to refer to specific amino acids as they are discussed in the text.*

Residue numbers have been added to the antibody sequences in Supplementary Fig. 4.

10. *Supplementary Fig. 5. A scale bar is missing from the viral lineage tree (unless it's not drawn to scale, in which case that should be indicated).*

A scale bar has been added with the viral Env lineage tree in Supplementary Fig. S6 (formerly Fig. S5). We also indicate in the caption that the units of the scale bar are "nucleotide substitutions per site". We thank the reviewer for pointing out this omission.

11. *Page 6 – The sentence "Overall, H4K3 is exceptionally broad and equipotent with 4E10 that is protective against HIV-1 in animal models" is confusing and I am not sure what the reference to animal models is referring to.*

We apologize for the confusion. We refer to a study by Hessel et al., in which 4E10 protected against mucosal challenge by SHIV_{Ba-L} in Indian rhesus macaques, which is a relevant animal model of HIV infection. Since H4K3 shows such similarity in *in vitro* activities with 4E10, it might exhibit similar protective properties. The text in question (Page 6) has been re-written for clarity.

12. *Page 7 – ADA-CM needs to be defined.*

We have now provided the identity of the seven mutations in ADA (actually, there were 9 mutations in total as some mutations originally occurred in pairs) that comprise the ADA-CM mutant (Page 7). We also reference the paper that first described this highly stable membrane-embedded Env trimer.

13. Were other positions mutated to Ala other than the ones mentioned on page 10? If so, it should be indicated (even if those mutations had no effect).

An MPER Ala-scan panel of COT6 virus mutants was used in neutralization assays with PGZL1, H4K3 and 4E10, the results of which are shown as bar graphs in Fig. S6; the results and this figure are also referred to in the text. The most notable and consistent Ala resistance mutations were W772, F673, D674, L679 and W680. We now also discuss (on Page 11) the effects of other MPER Ala mutations in this part of the manuscript. Notably, mutations L663A, D664A, S665A, W666A, K667A, L669A, W670A, K677A and W678A all enhanced neutralization by the three MPER bnAbs, as previously reported for 4E10. The neutralization enhancement effect with these mutants is not fully understood, but might be explained by changes in Env conformation that increase accessibility and/or susceptibility to functional inactivation by the MPER bnAbs.

14. On page 12, before the section on the cryo-EM structure, it should be indicated that the model was obtained using MD simulations. It wasn't clear until the following results section. Also, where is K683 in the corresponding figure (panel 5G)?

The text in question has now been updated to indicate that our model was obtained using MD simulations. In addition, the location of K683 is now indicated on Fig. 5, panel G.

15. Page 13 – the sentence should read “Through crystallography and cryo-EM, we have determined the orientation, angle of approach ...”

The sentence in question has now been amended.

We sincerely thank Reviewer#2 for the helpful comments.

Reviewer #3

Zhang and colleagues report on a novel, broadly neutralising MPER antibody (PGZL1) that shares many features with mAb 4E10. Importantly, PGZL1 shares germline V/D-region genes with 4E10 (V-region genes VH1-69 and VK3-20, as well as D-region D3-10), but has a shorter CDRH3 and lower poly-reactivity. To provide insights into how PGZL1 variants interact with MPER on the viral membrane, the authors conducted a detailed structural analysis of lipid-bound PGZL1 variants and performed cryo-EM reconstruction of Env AMC011 in complex with PGZL1.

The authors highlight several notable features of PGZL1: (i) PGLZ1 exhibits high breadth (84% neutralisation against a panel of 130 different viruses; 100% for variant H4K3); (ii) PGZL1 is intriguingly an IgG1, in contrast to 4E10 and many other MPER neutralising antibodies (which were isolated as IgG3); and (iii) the PGZL1 CDRH3 is three residues shorter, and uses a different J gene, compared to 4E10. The authors also emphasise that a

PGZL1 germline revertant (that contains the mature PGZL1 CDR3s) still retains considerable neutralising activity.

We appreciate these supportive comments. We address Reviewer #3's comments below.

- 1. The study is mostly very thorough and elegantly done. It would, however, be a beautiful paper if it were not for a severe omission. The authors present PGZL1 as if it would be the very first 4E10-like antibody that has been discovered. In truth, Krebs et al (Immunity 2019; DOI:10.1016/j.immuni.2019.02.008) recently reported on three lineages of MPER antibodies in a single donor. The lead antibody discovered by Krebs et al (VRC42) is also an IgG1 MPER-specific Ab that uses VH1-69 and VK3-20. The original VRC 42 clone appears to have a higher breadth than the original PGZL1. Similar to what is seen for PGZL1, Krebs et al. also report that a germline revertant of VRC34 retains the ability to engage the MPER antigen. Thus, all key findings reported here by Zhang et al. with PGZL1 have also been observed for VRC34. It is not easy to comprehend why, knowing of the competing study (Krebs et al. was published online March 12), the authors did not adapt their manuscript before submission. Given the similarities between PGZL1 and VRC42, analysis of PGZL1 must also include VRC42. This includes all parts of the manuscript, sequence alignments (Fig S1), phylogenetic analyses (Fig 1c), and Table S2 to name but a few. The entire introduction and discussion must also be adapted to highlight where PGZL1 may be superior to VRC42 and/or where the analysis of PGZL1 adds to the understanding of MPER reactive antibodies that goes beyond that shown by Krebs et al. Without these points being addressed, the importance of the findings by Zhang and colleagues cannot be rated.*

We regret the impression that our antibody would be the first 4E10-like antibody. This was an unintended outcome of being under pressure to publish our results in a timely manner. However, we are pleased to now include a comparison of PGZL1 antibodies and antibody VRC42 described by Krebs et al., which was not published when we began to write the paper. Our revised paper now includes relevant updates to the introduction and discussion sections, as well as to phylogenetic analysis (Page 4, Figure 1C), sequence alignment (Page 4-5 and 8, Figure S1A), hydrophobicity plots (Page 13, Figure S1B), gene usage (Page 4-5, Table S2) and structure (Page 10, Figure S5). As we iterated in our answer to a comment 1 from Reviewer #2, these comparisons strengthen our manuscript by showing: (i) how these 4E10-like antibodies evolved from similar V_H , V_K and D_H germline genes and three different J-region germline genes into antibodies with differences in V_H/V_K SHMs that can perform a very similar function; (ii) 4E10-like antibodies contain hydrophobic residues in similar positions in the CDRH3 with (iii) some similar residues in N2 addition regions in the CDRH3; (iv) structural comparison reveals slight differences in the binding of the N-term of the MPER epitope (residues 671-675) perhaps due to different residues from the different J gene usage (F100g in PGZL1, H4K3 and 4E10 and M100g in VRC42.01).

As an advance over the Krebs et al. study, our X-ray structural analysis of PGZL1 and H4K3 in complex with lipids, identifies lipid binding sites on these antibodies that together with cryo-EM structural data allow us to infer the orientation and the angle of approach to the composite epitope (MPER-viral membrane), which is key information for immunogen design and vaccine development, as also noted by Reviewer #1.

- 2. Patient/sampling information is missing. For how long has the donor been infected? Were the 3 time points used for NGS before or after the date of Ab isolation?*

We kindly refer to our reply to a similar question (#2 from Reviewer #2). To summarize here, patient sampling date information has now been included in Page 4-6 in addition to Figure 1D in the initial manuscript. However, the time of infection with this donor is not available. PGZL1 was isolated by single cell sorting from sample collected on visit 5 (04/15/09) and BCR repertoire NGS sequence analysis was performed on sample collected on Visit 2 (11/18/08), Visit 4 (03/12/09) and Visit 6 (05/18/09). The time points for NGS analysis were from Visit 2 and Visit 4 (prior to the PGZL1 isolation visit) and Visit 6 (after PGZL1 isolation visit).

3. *Fig. 2F: Why is breadth at 200 ug/ml reported, as opposed to at 50ug/ml in Fig 1?*

We reported the neutralization breadth of PGZL1 gVmDmJ at 200 ug/ml because it shows that this antibody, although lacking in potency, does recognize and neutralize a significant fraction of HIV-1 isolates, which is very different information for vaccine design purposes than if it only neutralized a few isolates even at a high concentration. However, the breadth at 50 ug/ml is a typically used cut-off for neutralization breadth, so the table has now been modified to include values at 50 ug/ml instead.

4. *“The IC50s of PGZL1 and H4K3 are tightly correlated ($r=0.82$, $p<0.0001$) suggesting a similar neutralization mechanism (Figure S3C).” This refers to fig S3D not S3C.*

The figure reference has been corrected. We thank the reviewer for pointing out the typo.

5. *“The IC50s of PGZL1 and PGZL1 gVmDmJ correlated modestly ($r=0.52$, $p=0.0013$) and some isolates were hypersensitive to the revertant (Figure S3C).” What do the authors define as hypersensitive? Table S5 does not show any virus with lower IC50 against the revertant than the wt Ab.*

We apologize for the confusion- we used the word ‘hypersensitive’ incorrectly. Our intention was to make a point about an imperfect correlation between IC₅₀s of mature and revertant PGZL1 antibodies. Such variation in the IC₅₀ ratio between isolates may reveal information about the relative sensitivity of a particular Env for revertant vs wildtype PGZL1. We wanted to point out that there are some viruses that were *relatively* sensitive (though granted not really hypersensitive) to the PGZL1 revertant, which might be useful information in selecting Env immunogens intended to elicit PGZL1-like antibodies. We have therefore re-worded the text.

6. *Since the authors did not succeed in isolating infection competent Env clones from the PGZL1 donor, the data on PG13 Env and the putative resistance mutation should probably be dropped. This does not add to the paper without functional confirmation. Sticking with the COT6 data is cleaner.*

While we agree that the COT6 neutralization data are cleaner, we believe that the sequences of contemporaneous Envs from Donor PG13 in Fig. S6 provide a valid and complementary data set. In fact, these Env sequences rescued from the PGZL1 donor revealed the presence of polymorphisms in the MPER (e.g. S674, E674 and T674), which were introduced into COT6 and were subsequently shown in Fig. 4 to cause relative resistance (7.7 to 120-fold increase in IC₅₀) to PGZL1. We note that the other reviewers did not express concern about including these Env sequences.

7. *The authors state in the abstract that PGZL1 lacks polyreactivity, but conclude in the context of Figure 3: “Overall, PGZL1 antibodies showed less polyreactivity than 4E10”. Less certainly does not mean lacking. This needs to be corrected in the abstract.*

We agree and have substituted “lacks polyreactivity’ in the abstract to ‘is less polyreactive’.

8. *More data on the patient plasma would have been nice to see. Does PGZL1 capture the breadth of the plasma?*

As stated above, patient plasma was very limited, which precluded many polyclonal antibody-mapping experiments, e.g. mass-spec and cryo-EM analyses using plasma-derived antibodies that might have provided insight. We also acknowledge that the IC₅₀s of bnAbs isolated by single B cell sorting may not always correlate with that of cognate donor plasma due to an underrepresentation of plasma antibodies similar to the bnAb isolated from B cell sorting and/or to the presence of an abundance of other antibodies with distinct HIV-1 neutralizing activity in the plasma. (Of note, 10E8 and cognate donor plasma did not correlate well, p=0.11; Huang et al., *Nature* 2012). However, we did present plasma neutralization and binding data showing that 4E10-like bnAbs were present in the PG13 donor at Visit 2, which was prior to the Visit 5 sample that was used for single B cell sorting and antibody cloning. This information, although not as comprehensive as we would have liked, nevertheless provides important support for our much more comprehensive molecular analyses on the PGZL1 monoclonal bnAbs, and verify that PGZL1-like antibodies were present in the plasma of this patient.

9. *The data on the germ line revertant are difficult to interpret as not all variants have been tested in all assays. In Fig 2B, PGZL1gVgDgJ and 4E10 gVgDgJ need to be shown. All three variants of 4E10 (wt., 4E10 gVmDMJ and 4E10 gVgDgJ) need to be shown in Fig 2C. 4E10 gVmDmJ_L100cF, as it has improved neutralisation capacity according to Figure S4C, should also be included in these binding experiments.*

Binding experiments on the PGZL1 and 4E10 germline revertants PGZL1 gVgDgJ, 4E10 wt, gVgDgJ and gVmDmJ have now been included in the Figs. 2B and 2C. These represent a substantial number of new ELISA and BLI experiments that were performed using the same antibody stocks between the two binding assays. The results show good agreement between ELISA and BLI, and add support to our original conclusion that PGZL1 gVgDgJ binds to MPER peptide (KD=938 nM) where the 4E10 gVgDgJ counterpart does not bind. We further find that PGZL1 gVmDmJ binds to MPER peptide (KD=64nM) with greater affinity than the 4E10 counterpart (KD=311nM). These binding data also align and help with the overall interpretation of our neutralization results.

Reviewers' Comments:

Reviewer #1:

Remarks to the Author:

The authors have addressed the concerns of the reviewers largely in the revised manuscript.

Reviewer #2:

Remarks to the Author:

Zhang et al. have revised their manuscript on an MPER-directed bnAb, called PGZL1, and its variants, from an HIV-1 infected patient and have now included a comparison to the recently published VRC42 MPER-directed bnAb. These are also antibodies of the major IgG1 class and are less polyreactive than the 4E10 MPER bnAb. Their comparisons highlight how different individuals have produced similarly functioning MPER bnAbs from different J-region genes and similar VD genes that evolved differently. The fact that such bnAbs could be produced in multiple instances by different pathways is promising for immunogen design. The biochemical and structural data presented also provide evidence for lipid binding, which is a useful addition for immunogen design purposes.

Major comments/concerns/questions have been satisfactorily addressed. There are still some minor adjustments that should be made, as indicated below.

1. Should BJOX025 be changed to BJOX025000 in Figures 2G and S4C? What about for BJOX028?
2. The suggestion that H4K3 might offer protection against SHIV challenge in monkeys is still unclear. I think the point would be made more clear if the last sentence of the "Broad HIV neutralization by PGZL1 lineage antibodies" is changed to "Overall, H4K3 is exceptionally broad and equipotent with 4E10, which has been reported to protect against SHIV challenge in monkeys. Thus, H4K3 might exhibit similar protective properties". Alternatively, the latter part of the sentence should be omitted or mentioned in the discussion.
3. Figure 6 C-E should have more descriptive titles for readability. In C, it should be clear that BLI data refers to binding to MPER; D should indicate it's against membrane. The same should be included in E. Additionally, "Octet binding" text in E should be replaced by "BLI".

Reviewer #3:

Remarks to the Author:

all my queries have been adressed

Point-by-point response to issues raised by referees (NCOMMS-19-09017)

We thank the reviewers for re-reviewing the manuscript and for their final comments.

Reviewer #1 (Remarks to the Author):

The authors have addressed the concerns of the reviewers largely in the revised manuscript.

We are delighted that Reviewer #1 approved our revisions. We thank the reviewer for their input.

Reviewer #2 (Remarks to the Author):

Zhang et al. have revised their manuscript on an MPER-directed bnAb, called PGZL1, and its variants, from an HIV-1 infected patient and have now included a comparison to the recently published VRC42 MPER-directed bnAb. These are also antibodies of the major IgG1 class and are less polyreactive than the 4E10 MPER bnAb. Their comparisons highlight how different individuals have produced similarly functioning MPER bnAbs from different J-region genes and similar VD genes that evolved differently. The fact that such bnAbs could be produced in multiple instances by different pathways is promising for immunogen design. The biochemical and structural data presented also provide evidence for lipid binding, which is a useful addition for immunogen design purposes.

Major comments/concerns/questions have been satisfactorily addressed. There are still some minor adjustments that should be made, as indicated below.

We thank Reviewer #2 for the positive comments. Below we address the remaining minor comments.

1. Should BJOX025 be changed to BJOX025000 in Figures 2G and S4C? What about for BJOX028?

BJOX025 and BJOX028 have been corrected to BJOX025000 and BJOX028000 in Figs. 2G and S4C.

2. The suggestion that H4K3 might offer protection against SHIV challenge in monkeys is still unclear. I think the point would be made more clear if the last sentence of the “Broad HIV neutralization by PGZL1 lineage antibodies” is changed to “Overall, H4K3 is exceptionally broad and equipotent with 4E10, which has been reported to protect against SHIV challenge in monkeys. Thus, H4K3 might exhibit similar protective properties”. Alternatively, the latter part of the sentence should be omitted or mentioned in the discussion.

As suggested by the reviewer, the sentence in question has been modified to: "Overall, H4K3 is exceptionally broad and equipotent with 4E10, which has been reported to protect against SHIV challenge in monkeys."

3. Figure 6 C-E should have more descriptive titles for readability. In C, it should be clear that BLI data refers to binding to MPER; D should indicate it's against membrane. The same should be included in E. Additionally, "Octet binding" text in E should be replaced by "BLI".

We edited the legend of Figure 6 C-E and Figure 6E and included the requested modifications. We thank Reviewer #2 for their comments.

Reviewer #3 (Remarks to the Author):

all my queries have been addressed

We much appreciate that Reviewer #3 approved of the revised manuscript, and we thank the reviewer for their time and effort.